# Decoupling of respiration rates and abundance in marine prokaryoplankton

Jacob H. Munson-McGee[1,6], Melody R. Lindsay[1,6], Eva Sintes[2,3], Julia M. Brown[1], Timothy D'Angelo[1], Joe Brown[1], Laura C. Lubelczyk[1], Paxton Tomko[4], David Emerson[1], Beth N. Orcutt[1], Nicole J. Poulton[1], Gerhard J. Herndl[2,5] & Ramunas Stepanauskas[1✉]

The ocean–atmosphere exchange of $CO_2$ largely depends on the balance between marine microbial photosynthesis and respiration. Despite vast taxonomic and metabolic diversity among marine planktonic bacteria and archaea (prokaryoplankton)[1–3], their respiration usually is measured in bulk and treated as a 'black box' in global biogeochemical models[4]; this limits the mechanistic understanding of the global carbon cycle. Here, using a technology for integrated phenotype analyses and genomic sequencing of individual microbial cells, we show that cell-specific respiration rates differ by more than 1,000× among prokaryoplankton genera. The majority of respiration was found to be performed by minority members of prokaryoplankton (including the *Roseobacter* cluster), whereas cells of the most prevalent lineages (including *Pelagibacter* and SAR86) had extremely low respiration rates. The decoupling of respiration rates from abundance among lineages, elevated counts of proteorhodopsin transcripts in *Pelagibacter* and SAR86 cells and elevated respiration of SAR86 at night indicate that proteorhodopsin-based phototrophy[3,5–7] probably constitutes an important source of energy to prokaryoplankton and may increase growth efficiency. These findings suggest that the dependence of prokaryoplankton on respiration and remineralization of phytoplankton-derived organic carbon into $CO_2$ for its energy demands and growth may be lower than commonly assumed and variable among lineages.

The black-box approach to prokaryoplankton respiration presents a stark contrast to the overwhelming evidence of prokaryoplankton's considerable phylogenetic and genomic diversity[1–3] and vast differences in growth and organic substrate uptake rates among lineages, as indicated by microautoradiography fluorescence in situ hybridization (MAR-FISH)[8], FISH-nanoscale secondary ion mass spectrometry (nanoSIMS)[9] and stable isotope probing (SIP)[10]. Some of the genome-predicted metabolic processes, such as proteorhodopsin-based phototrophy[3,5,6], may have a direct effect on prokaryoplankton respiration and $CO_2$ release to the atmosphere, but their global importance remains poorly constrained. Here we developed a method for integrated in situ oxygen respiration rate measurements and genomic sequencing of individual microbial cells to show that respiration rates differ by more than three orders of magnitude among prokaryoplankton genera. Our results provide evidence for the importance of proteorhodopsin phototrophy as a complementary energy source to respiration in prevalent prokaryoplankton lineages and its potential impact on the global carbon cycle. These findings demonstrate the feasibility of directly linking microbial genomes and phenomes at the single-cell resolution and emphasize the importance of breaking the marine prokaryoplankton black box into functionally more meaningful components in ecosystem models.

## Single-cell respiration calibration

RedoxSensor Green (RSG) has previously been used in laboratory and environmental studies of diverse microorganisms as a cell viability probe specific to oxidoreductase activity[11]. To evaluate the feasibility of using RSG in a quantitative manner, we analysed the relationship between bulk oxygen consumption and single-cell RSG fluorescence in pure cultures of phylogenetically diverse, aquatic bacteria (the methodological workflow is illustrated in Extended Data Fig. 1 and Supplementary Table 1). Stationary-phase cells varied in RSG fluorescence intensity in all cultures (Extended Data Fig. 2a), indicating physiological heterogeneity consistent with previous studies[12,13], but were distinct from the negative controls (Extended Data Fig. 2b). The average per-cell fluorescence correlated with the average per-cell rate of oxygen consumption within the analysed range (around 1–1,000 amol $O_2$ per cell per h, $R^2 = 0.86$), with no evidence of taxonomic biases (Fig. 1a). This culture calibration enabled the use of RSG fluorescence intensity as a proxy for an individual cell's respiration rate.

To validate the use of RSG in quantifying respiration in environmental studies, we performed simultaneous single-cell RSG probing and bulk respiration measurements using the traditional Winkler methodology[14] on geographically diverse prokaryoplankton samples under

[1]Bigelow Laboratory for Ocean Sciences, East Boothbay, ME, USA. [2]Department of Functional and Evolutionary Ecology, University of Vienna, Vienna, Austria. [3]Instituto Español de Oceanografía-CSIC, Centro Oceanográfico de Baleares, Palma, Spain. [4]Purdue University, West Lafayette, IN, USA. [5]Department of Marine Microbiology and Biogeochemistry, Royal Netherlands Institute for Sea Research (NIOZ), Utrecht University, Den Burg, The Netherlands. [6]These authors contributed equally: Jacob H. Munson-McGee, Melody R. Lindsay. ✉e-mail: rstepanauskas@bigelow.org

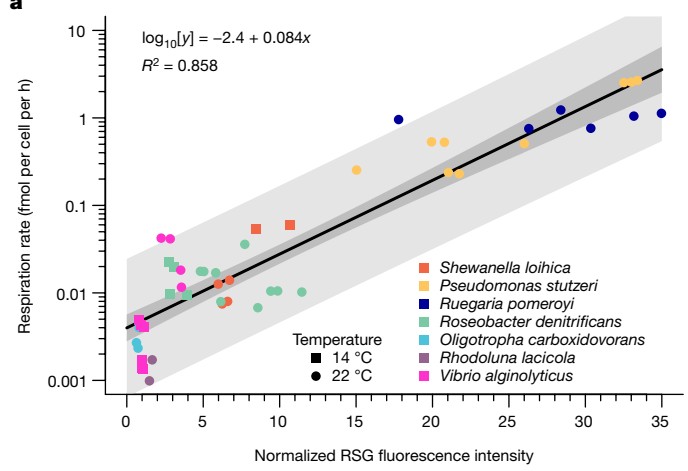

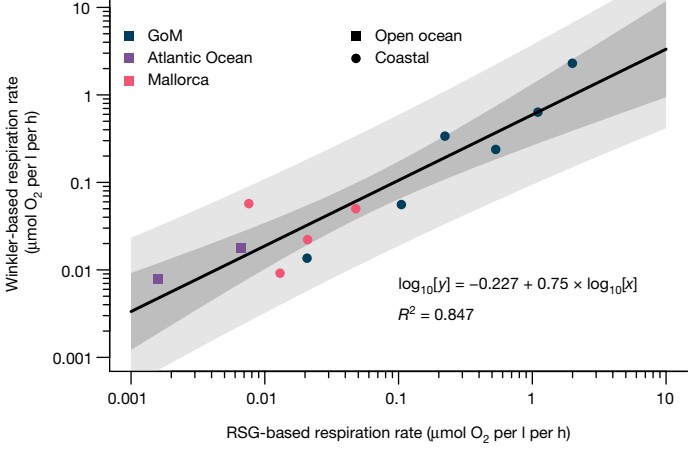

**Fig. 1 | Calibration and validation of RSG-based prokaryoplankton respiration measurements. a**, Comparison of normalized RSG fluorescence intensity with cell-specific $O_2$ consumption rates in laboratory cultures. Each point represents one culture experiment ($n = 50$), with the fluorescence values normalized to fluorescence bead standards used to intercalibrate the instrument. Additional experimental details are provided in Supplementary Table 1. **b**, Comparison of bulk marine prokaryoplankton respiration rates

determined using Winkler and cell-specific RSG methods ($n = 12$). For the RSG method, the per-cell average respiration rate was calculated from per-cell respiration rates and then used to calculate the total amount of $O_2$ consumed together with the number of RSG labelled cells per ml. In both plots, the darker-shaded area represents the 95% confidence interval, and the lighter-shaded area is the 95% predicted interval, around the regression line.

in situ conditions. Cell-specific $O_2$ consumption rates were calculated using the culture calibration (Fig. 1a) and were used to estimate bulk consumption rates. Both techniques produced similar estimates of $O_2$ consumption throughout the analysed range of 0.008–2.3 µmol $O_2$ per l per h (Fig. 1b). This provided further evidence that RSG can be used as a proxy for microbial oxygen respiration rates in an environmental context. On the basis of a variety of killed control samples, the cell-specific analytical detection threshold was found to be around 4 amol $O_2$ per cell per h (Extended Data Fig. 3a,b) in environmental samples. The lowest cell abundance that we could quantify using our flow cytometry analysis was around $10^6$ cells per l (ref. [15]). This translates to a theoretical detection limit of RSG-based microbial community respiration measurements of around 4 pmol $O_2$ per l per h, which is about four orders of magnitude lower than the standard $O_2$ consumption assays using Winkler or optode sensor detection[14]. Thus, the RSG approach may enable microbial respiration measurements in low-biomass and/or low-activity environments, in which the current methods lack sensitivity. Furthermore, RSG probing takes only minutes, which increases the temporal resolution of the measurement and reduces the risk of bottle effects compared with the typically ≥24-h-long incubations for Winkler measurements[4]. Importantly, this approach uniquely measures respiration at the resolution of individual cells and in a non-destructive manner, which offers opportunities to study variability in intercellular respiration and combine it with other analyses of the same cells.

## Single-cell genome and phenome analyses

To compare respiration rates among prokaryoplankton lineages and relate those measurements to the genome content and cell size of the lineages, we performed RSG probing and subsequent single-cell genomics on six marine prokaryoplankton samples collected from the coastal Gulf of Maine (GoM; 43.86° N, 69.58° W) over a period of two years (Supplementary Table 2). Single amplified genomes (SAGs) were generated and sequenced using previously described techniques[16], with the added use of RSG as a source of fluorescence in flow cytometry cell sorting, during which each cell's optical properties were recorded. Subsequently, the forward-angle light scatter intensity was used to estimate the diameter of each cell[16]. The cell-specific RSG fluorescence

in three of these samples, collected in October 2018, April 2019 and July 2019, was converted to respiration rates using the culture calibration described above. Owing to changes in instrument configuration, the three earlier samples, collected in 2017, were analysed in a similar manner but without the quantitative conversion of RSG fluorescence to respiration rates.

We recovered a similar fraction of genomes from individual cells that were probed with either RSG or the commonly used nucleic acid stain SYTO-9, demonstrating the compatibility of RSG with downstream single-cell DNA amplification and sequencing (Extended Data Fig. 3c). The RSG-based estimates of bulk respiration rates had a range of 0.35–0.78 µmol $O_2$ per l per h, consistent with previous measurements in the GoM[17] and other coastal environments[18]. The estimated respiration rates of individual cells spanned over five orders of magnitude, from below detection (<0.004 fmol $O_2$ per cell per h) to around 145 fmol $O_2$ per cell per h (Fig. 2a, Extended Data Fig. 4 and Supplementary Table 3). Although substantial variation in respiration rates occurred even among cells with nearly identical genomes, cells with a predicted average amino acid identity of above about 60% had more uniform respiration rates compared with cells with greater genome divergence, approximately corresponding to the boundary between genera in contemporary prokaryote taxonomy (Fig. 2b). Thus, to simplify our dataset in a biologically meaningful manner, we clustered SAGs at the genus level.

Our results revealed substantial differences in the cell-specific respiration rates and contributions to prokaryoplankton oxygen consumption among genera (Fig. 2c–e). On the high end of the spectrum, the genera ASP10-02a (Gammaproteobacteria), LFER01 and *Planktomarina* (both members of the *Roseobacter* clade, Rhodobacteraceae, Alphaproteobacteria) contained cells respiring more than 1 fmol $O_2$ per cell per h. *Planktomarina* contributed 10–37% of the total $O_2$ consumption at all three timepoints while comprising only 1.2–2.5% of cells. Cells with intermediate respiration rates of 0.01–1 fmol $O_2$ per cell per h were dominated by *Amylibacter* (*Roseobacter* clade), multiple genera of Bacteroidia, the Gammaproteobacteria genus *Thioglobus* and several less abundant genera (Extended Data Fig. 5). On the low end of the spectrum, 99 out of the 303 identified genera contained no cells exceeding our detection limit of around 0.004 fmol $O_2$ per cell per h. Notably, three out of the five most prevalent genera

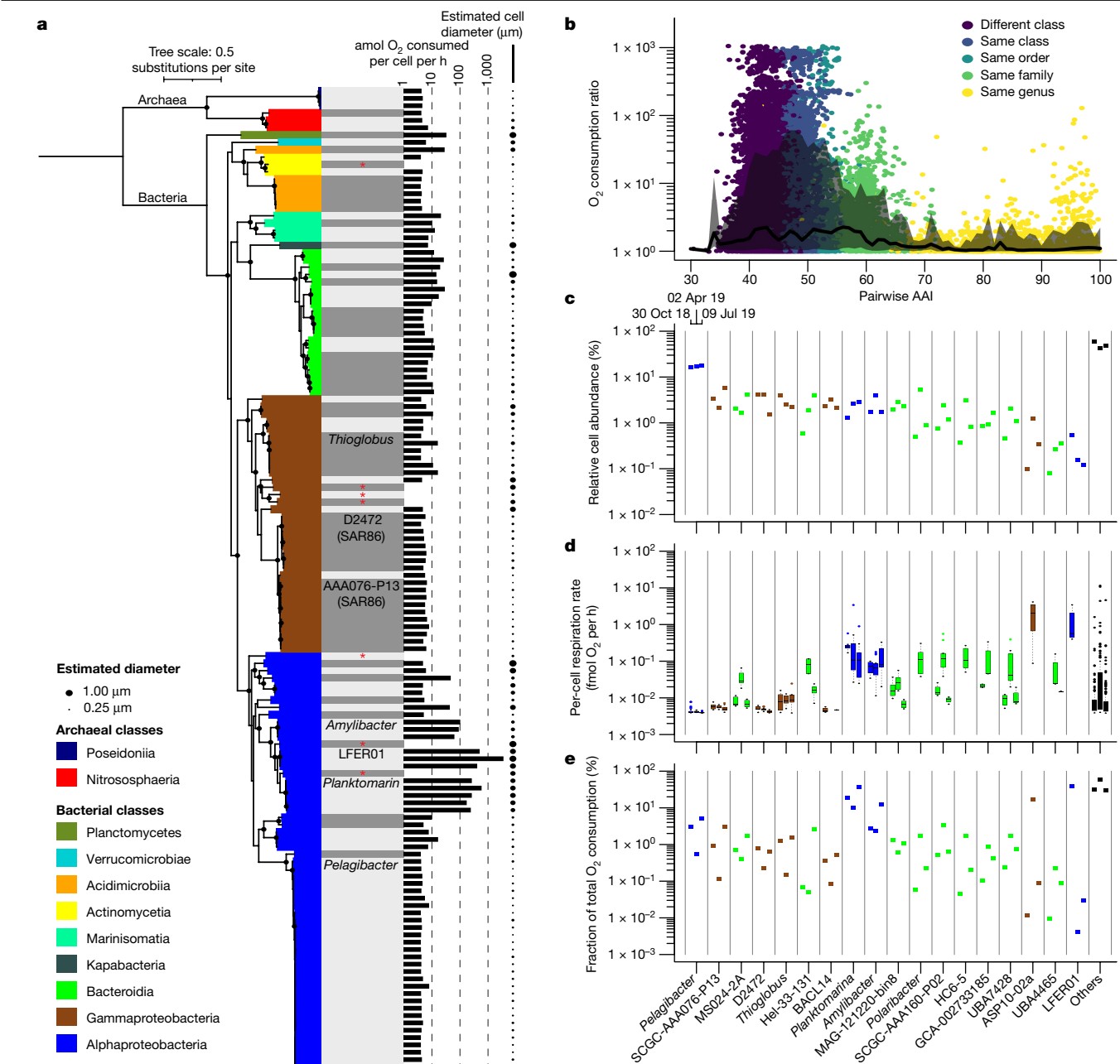

**Fig. 2 | Cell-specific respiration rates of GoM prokaryoplankton. a**, The phylogenetic composition, respiration rates and diameters of individual prokaryoplankton cells collected in October 2018. Maximum-likelihood tree of the 16S rRNA gene that was rooted in Archaea is shown. Bootstrap values of greater than 0.75 are indicated by black circles. Genera are separated by grey and white shading. The names of genera discussed in this Article are shown. Reference entries of cultured isolates are indicated by a red asterisk. **b**, The relationship between amino acid identity and the ratio of $O_2$ consumption in pairs of cells recovered from the same sample. The black line indicates the median $O_2$ consumption ratio at 1% intervals and the grey-shaded area is bounded by the 10% quantile below and the 90% quantile above. **c**, Genus-specific cell abundance on the basis of metagenomic read recruitment. **d**, Genus-specific per-cell respiration rates. The box and whisker plots indicate the median (centre line), first and third quartiles (box limits), and 1.5 × interquartile range (whiskers). The dots indicate outliers outside 1.5 × interquartile range. **e**, The fraction of prokaryoplankton respiration performed by each genus. Included in **c**–**e** are the ten most abundant genera and ten genera with the greatest contribution to community respiration. The vertical grey lines separate genera. The colours in **c**–**e** are as defined in **a**. Note that the genus-specific respiration estimates take into account cells that fall below the RSG detection limit, as described in the Methods. For **c**–**e**, the sampling date is indicated in the top left of **c** and the number of sorted and sequenced cells per day was $n = 282$ (30 October 2018), $n = 274$ (2 April 2019) and $n = 277$ (9 July 2019).

contained no cells respiring more than 0.03 fmol $O_2$ per cell per h at any timepoint: the Alphaproteobacteria genus *Pelagibacter* and two genera of the Gammaproteobacteria cluster SAR86 (AAA076-P13 and D2472). *Pelagibacter* and SAR86 collectively constituted 22–24% of all cells across the timepoints, while accounting only for 0.6–5.6% of the total prokaryoplankton respiration. The RSG-based estimate

of the average $O_2$ consumption by *Pelagibacter* in GoM (around 5.0 amol $O_2$ per cell per h) was similar to previous measurements in a stationary-phase culture (around 1–10 amol $O_2$ per cell per h)[19]. Similarly, the *Roseobacter* clade and ASP10-02a have been reported to be over-represented among the most active prokaryoplankton cells in algal blooms and coastal environments[20–22]. Thus, our findings are in

general agreement with previous studies, suggesting that RSG-based single-cell analyses offer a realistic assessment of the differences in respiration rates among prokaryoplankton taxa.

When considering the three experiments in which cell-specific respiration was quantified, the most pronounced seasonal differences were observed in the gammaproteobacterial genus ASP10-02a and various genera of Flavobacteriaceae (Bacteroidia) (Fig. 2c,d). These genera had the highest cell-specific respiration rates and cell abundances in April 2019. Similarly, Flavobacteriaceae and ASP10-02a had elevated RSG fluorescence and relative cell abundance in April compared with during August and November, in the non-calibrated experiments of 2017 (Extended Data Fig. 6). Although this study has limited temporal resolution, it is noteworthy that the two April experiments took place during or shortly after GoM spring algal blooms (Extended Data Fig. 3d). Our findings are consistent with previous reports of both Flavobacteriaceae[23–26] and ASP10-02a[21] being associated with algal blooms. Notably, most other prokaryoplankton genera did not have pronounced differences in cell-specific respiration rates among sampling dates, despite the large seasonal differences in temperature and primary productivity in GoM[27].

We performed similar single-cell respiration and genomic analyses of samples from geographically diverse, offshore locations in the Atlantic and Pacific Oceans (Supplementary Table 2). On average, microorganisms from these oligotrophic environments had lower per-cell respiration rates compared with prokaryoplankton from the GoM. Similar to GoM, many of the most abundant lineages, including *Pelagibacter* and SAR86, had extremely low respiration rates (<10 amol $O_2$ per cell per h), while most of the $O_2$ consumption could be attributed to low-abundance genera (Extended Data Figs. 7 and 8). Collectively, these and GoM results suggest a general decoupling of respiration rates and cell abundance among marine prokaryoplankton taxa.

With the exception of Cyanobacteria, heterotrophy coupled to oxygen respiration is generally assumed to be the predominant source of energy for surface ocean prokaryoplankton. Thus, our finding that some of the most abundant prokaryoplankton lineages, in particular *Pelagibacter* and SAR86, have extremely low in situ cell-specific respiration rates is puzzling. This raises questions about how these microorganisms maintain numeric predominance and are not outcompeted by genera with around 100× higher in situ cell-specific respiration rates, such as *Planktomarina* and ASP10-02a, and how these findings can be reconciled with previous reports of relatively high in situ rates of cell doubling, protein synthesis and substrate uptake by *Pelagibacter*[28].

## Insights into the respiration paradox

Our finding of consistently low respiration rates among all analysed *Pelagibacter* and SAR86 cells across all sample collection dates and locations (Fig. 2 and Extended Data Figs. 4, 7, 8 and 10) contradicts previous hypotheses of rampant growth by some ecotypes of *Pelagibacter* that are fine-tuned to local conditions[29]. Small cell size may provide a partial explanation for their minimal respiration. Cells of genera *Pelagibacter*, AAA076-P13 and D2472 averaged around 0.3 μm in diameter, whereas the diameter of *Planktomarina*, *Amylibacter*, LFER01 and ASP10-02a averaged 0.5–0.8 μm (Supplementary Table 4). Assuming a similar cell shape, a twofold difference in diameter translates to an eightfold difference in biovolume. This is substantial, but not sufficient to explain the much greater discrepancies in cell-specific respiration rates. Furthermore, we observed only a weak correlation between a cell's volume and respiration rate (Fig. 3a). Instead, the RSG-based respiration rates correlated more strongly with genome-predicted minimal doubling time[30] (Fig. 3b), suggesting that growth rates of many genera were close to their theoretical maximum in the analysed GoM samples. Interestingly, we found no correlation between respiration and the frequency of cells containing viral DNA among genera, nor differences in SAR86 respiration rates between dates with high (October 2018) and low (July 2019) fraction of virus-containing cells (Extended Data Fig. 3e), which argues against phage–host interactions having a major impact on cell-specific prokaryoplankton respiration rates.

To gain additional insights into the physiology of GoM prokaryoplankton, we generated quantitative metatranscriptomic datasets[31] from four of the analysed samples. Individual reads were annotated using GoM SAGs as a reference database[3] and normalized to transcripts per cell. The overall average counts of mRNA and 16S rRNA gene transcripts per cell were 36 and 426, which is similar to the earlier reports on coastal prokaryoplankton[32]. However, our study showed that these counts varied widely among prokaryoplankton genera, ranging from 5 to 144 per cell for mRNA and from 48 to 2,480 per cell for rRNA. We observed a correlation between cell size and rRNA transcript copy number (Fig. 3c), but there were no significant correlations between cell-specific respiration rates versus counts of either rRNA or mRNA molecules (Fig. 3d,e), which are sometimes used as proxies for metabolic activity[33,34]. Furthermore, the per-biovolume counts of rRNA were relatively uniform among genera and differed by only threefold between *Pelagibacter* and *Planktomarina*. This suggests that ribosomal and total mRNA molecule counts are poor indicators of cellular respiration and that energy producing processes other than respiration may be important to the metabolism of GoM prokaryoplankton.

Proteorhodopsin-based phototrophy is a complementary source of ATP synthesis[35], the coding potential for which is widespread in marine prokaryoplankton[3,5–7], including most GoM genera (Supplementary Table 4). Thus, some prokaryoplankton lineages may be mixotrophs that rely on a combination of heterotrophic and phototrophic processes as energy sources. A recent study of pigment distribution in the Mediterranean Sea indicated that proteorhodopsin captures a similar amount of solar energy as chlorophyll *a*, which may be sufficient to sustain the basal metabolism of an average prokaryoplankton cell[36]. Although it is well known that *Pelagibacter* and other predominant groups of surface ocean prokaryoplankton consume simple organic compounds[20,28,37], access to proteorhodopsin-derived ATP may increase the use of these molecules in biosynthesis as opposed to respiration, therefore increasing growth efficiency[19,38,39]. However, direct evidence of the importance of proteorhodopsin in prokaryoplankton cellular energetics in situ and in the ocean's biogeochemical processes is lacking[5,6]. Here we found that the per-cell counts of proteorhodopsin transcripts were among the highest in taxa with some of the lowest detectable respiration rates, including *Pelagibacter* and SAR86, despite their small cell sizes (Fig. 4a). Given the extreme streamlining of *Pelagibacter* and SAR86 genomes and metabolism[37], the abundance of proteorhodopsin transcripts constitutes a strong indication of the importance of phototrophy to these predominant prokaryoplankton lineages.

Previous studies have demonstrated differential metabolism of prokaryoplankton between day and night, including a diurnal cycle of gene expression for transporters[40] and proteorhodopsins[41]. All of the RSG experiments described above were performed at mid-morning, when proteorhodopsin transcript counts may be the greatest[41]. To examine the potential impact of proteorhodopsin phototrophy on respiration rates, we conducted additional RSG probing and single-cell genomics experiments on GoM prokaryoplankton during the day and night in August 2021. We hypothesized that obligate heterotrophs increase their respiration rates during the daytime, when algal photosynthesis increases the availability of labile organic substrates. By contrast, non-cyanobacterial prokaryotic phototrophs may be expected to respire more at night to compensate for the absence of the proteorhodopsin-driven proton pump (Fig. 4b). Consistent with the first scenario, the genus UBA10364 (Bacteroidota, Flavobacteriales) exhibited higher respiration rates during the day compared with at night (Fig. 4c,d, Extended Data Figs. 9 and 10). By contrast, respiration rates of AA076-P13 (SAR86) were sixfold higher (52 versus 8 amol $O_2$ per cell per h) at night compared with during the day (Mann–Whitney *U*-test,

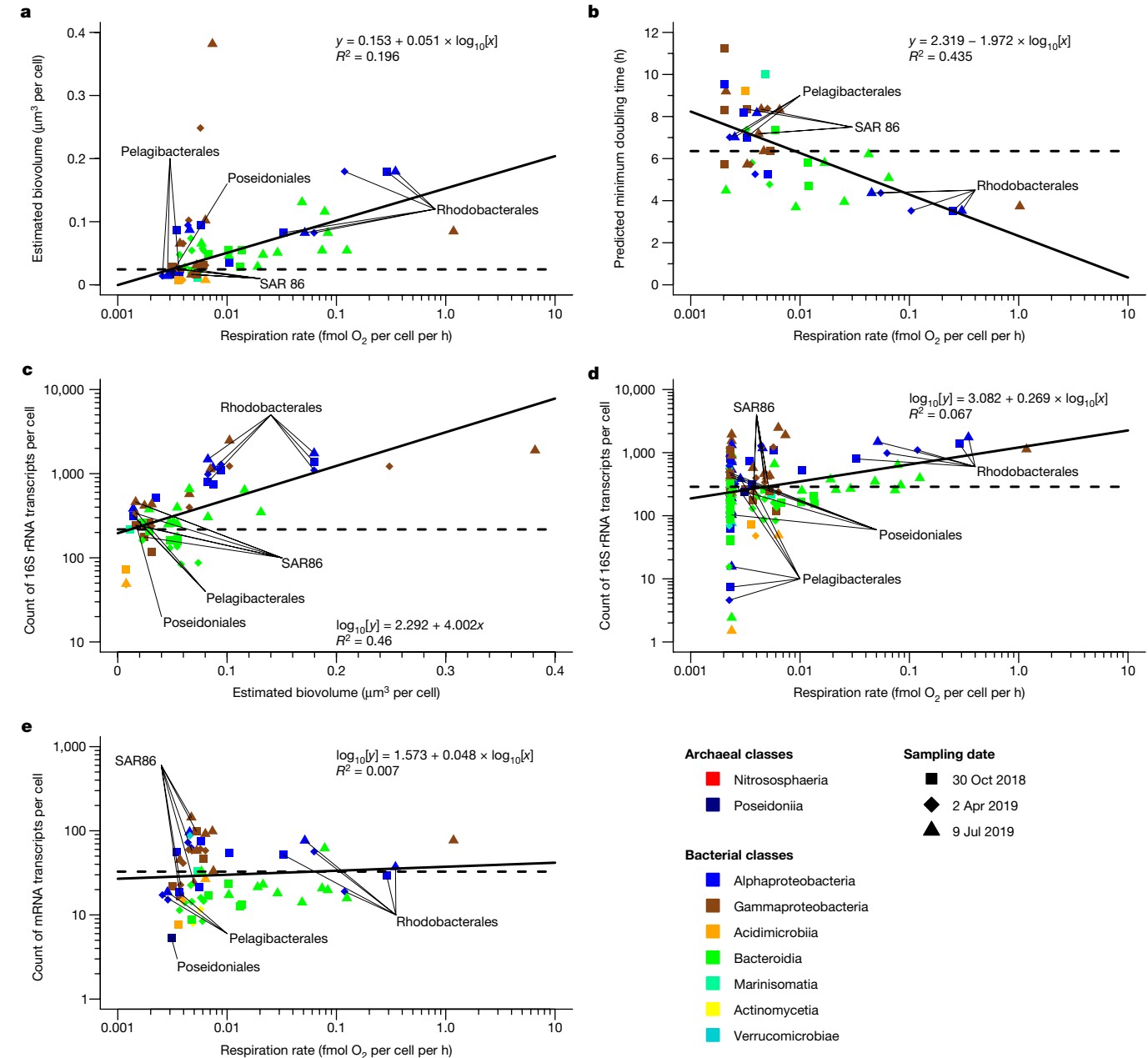

**Fig. 3 | Correlations among cell-specific prokaryoplankton properties.**
**a**, Weighted average $O_2$ consumption rate versus biovolume. **b**, Weighted average $O_2$ consumption rate versus minimal doubling time. **c**, Biovolume versus the count of 16S rRNA gene transcripts. **d**, Weighted average $O_2$ consumption rate versus the count of 16S rRNA gene transcripts. **e**, Weighted average $O_2$ consumption rate versus the count of mRNA transcripts. For **a**–**d**, each data point represents an average value calculated for each genus and field sample. The solid lines indicate regressions and the dotted lines indicate median values. Note that the weighted $O_2$ consumption rates take into account cells that fall below the RSG detection limit, as described in the Methods.

$P < 0.05$; Extended Data Figs. 9 and 10), suggesting a potential effect of phototrophy on the energy metabolism of this genus. SAR86 lineages have previously been shown to have a diverse set of proteorhodopsin genes[42,43], which they may use to supplement energy produced from respiration. The daytime reduction in AA076-P13 respiration by 44 amol $O_2$ per cell per h implies that a comparable amount of ATP may have been generated using the proteorhodopsin-driven proton gradient. However, the respiration of *Pelagibacter* was similar during the day and night, at around 5 amol $O_2$ per cell per h. Multiple factors may have masked the full impact of proteorhodopsin phototrophy on prokaryoplankton energy sources in our experiment, including (1) stimulation of respiration by photosynthesis products during daytime; (2) a limited ability of some genera to regulate their metabolism, including *Pelagibacter*[37]; (3) variable growth efficiency; and (4) insufficient

statistical power owing to the small scale of this experiment. By making a speculative assumption that the community-wide average impact of proteorhodopsin-driven proton pumps is equivalent to 44 amol $O_2$ per cell per h for 12 h a day, we can estimate that GoM prokaryoplankton are using proteorhodopsins to generate the ATP equivalent to that generated from 0.5 µmol $O_2$ per l per day, one order of magnitude less than the total prokaryoplankton respiration. The importance of proteorhodopsin phototrophy is expected to be much greater in oligotrophic, off-shore environments[38], in which respiration rates of less than 10 amol $O_2$ per cell per h are typical to most prokaryoplankton cells (Extended Data Figs. 7 and 8) and proteorhodopsin may absorb an amount of sunlight comparable to chlorophyll $a$[36]. In productive regions, the relatively constant access to proteorhodopsin-derived ATP may mitigate the impact of seasonal differences in phytoplankton

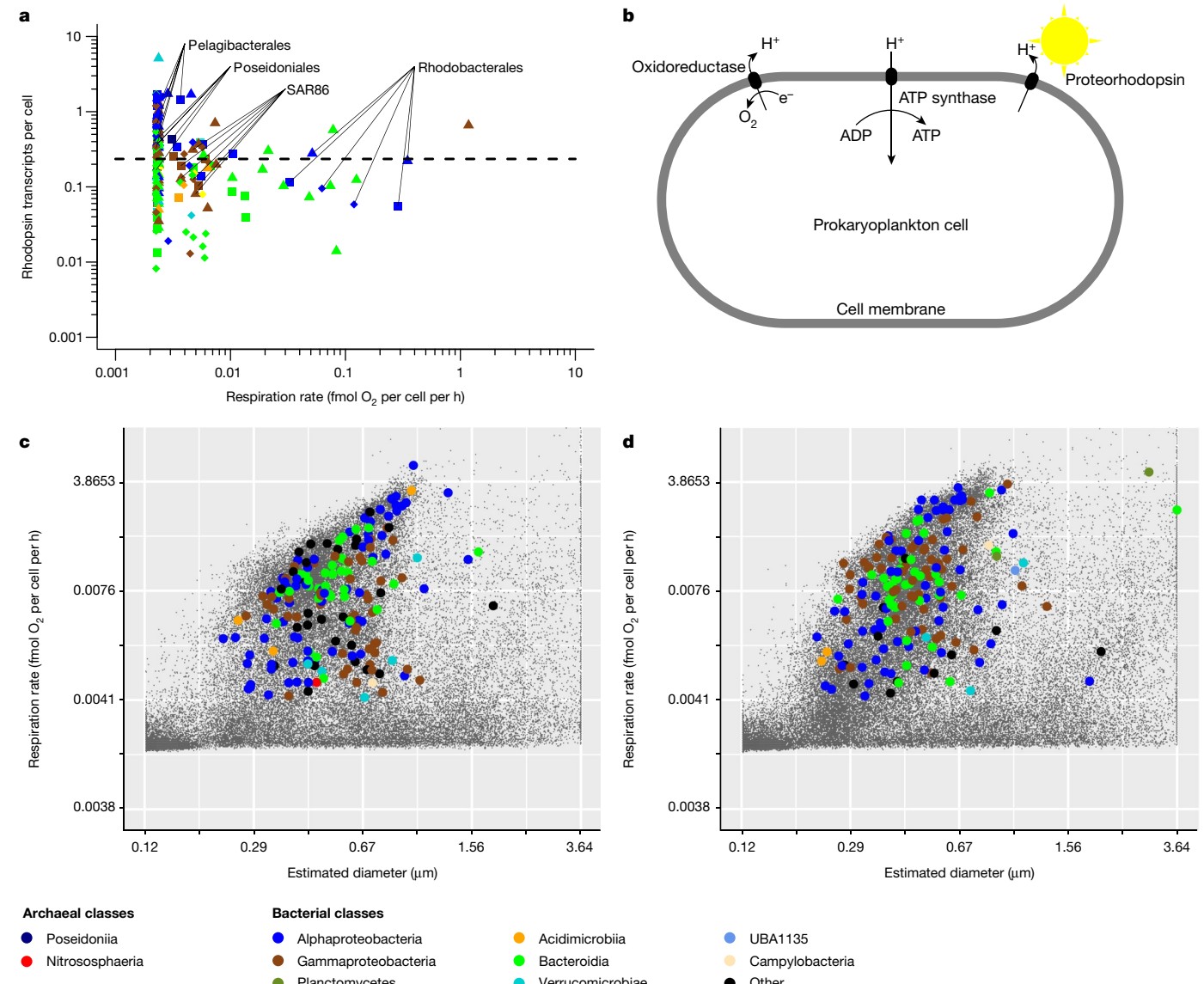

**Archaeal classes**
- ● Poseidoniia
- ● Nitrososphaeria

**Bacterial classes**
- ● Alphaproteobacteria
- ● Gammaproteobacteria
- ● Planctomycetes
- ● Acidimicrobiia
- ● Bacteroidia
- ● Verrucomicrobiae
- ● UBA1135
- ● Campylobacteria
- ● Other

**Fig. 4 | The potential impact of proteorhodopsin phototrophy on prokaryoplankton respiration. a**, The relationship between the cell-specific respiration rates and proteorhodopsin transcript counts among genera. Each data point represents a genus and a single field sample. Colours and shapes are as defined in Fig. 3. **b**, Schematic of the complementary roles of respiration and proteorhodopsin phototrophy in the generation of proton gradient and ATP in prokaryoplankton. **c**,**d**, Cell sizes, respiration rates and taxonomic affiliations of individual prokaryoplankton cells collected in GoM during August 2021 at night (**c**) and during the day (**d**). Note that the genus-specific respiration estimates take into account cells that fall below the RSG detection limit, as described in the Methods.

productivity on prokaryoplankton, helping to explain the limited seasonality in GoM prokaryoplankton composition (Fig. 2c–e and Extended Data Fig. 4).

may increase growth efficiency and productivity of the predominant prokaryoplankton groups, by either providing energy for the uptake of nutrients or driving the biosynthesis of biomolecules, resulting in a subsequent impact on carbon and energy fluxes in the surface ocean.

## Conclusions

Integrated genome and phenotype analyses of individual, uncultured cells revealed that cell-specific respiration rates differ by more than 1,000× among marine prokaryoplankton genera. We found that, although the majority of respiration is performed by relatively rare lineages, the most abundant prokaryoplankton taxa have extremely low respiration rates, calling into question the source of their competitive advantage. The decoupling of respiration rates and lineage abundance, the elevated counts of proteorhodopsin transcripts in low-respiring cells and the elevated respiration of some of the low-respiring taxa at night collectively indicate that proteorhodopsin-based phototrophy

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

## Methods

### Optimization of RSG incubation length

According to the manufacturer (Thermo Fisher Scientific), oxidoreductases convert RSG to a green fluorescent product inside live cells, which can be quantified using microscopy or flow cytometry. Compared with a functionally similar and previously utilized tetrazolium probe CTC[44], RSG has superior fluorescence spectral properties, a shorter incubation time and no known toxic effects[11]. The BacLight RedoxSensor Green Vitality Kit manual (Thermo Fisher Scientific) recommends a 10 min sample incubation with RSG before proceeding with cell analyses. To test the impact of incubation length on the abundance and fluorescence intensity of marine prokaryoplankton cells, we performed a test on seawater collected from coastal GoM (43.8609° N, 69.5782° W) on 11 August 2016. Two replicate surface seawater samples were collected in sterile 50 ml polypropylene tubes and transported to the lab at in situ temperature for further processing. These samples were amended with the manufacturer's recommended concentration of RSG (1 μM final concentration) and incubated at in situ temperature in the dark for a variable length of time before being analysed using the BD Influx Mariner flow cytometer (Becton Dickinson, formerly Cytopeia) equipped with a 488 nm laser for excitation and a 70 μm nozzle orifice. RSG-positive cells were gated on the basis of particle green fluorescence (531/40 bandpass), forward scatter and the ratio of green versus red fluorescence (692/40 bandpass; for improved discrimination of cells from detrital particles). We found that both the abundance and the fluorescence intensity of RSG-positive cells reached their maximum values and remained stable after 30–70 min of incubation (Extended Data Fig. 3f). On the basis of these results, subsequent RSG incubations were set for 30 min.

### Calibration of RSG fluorescence against O$_2$ respiration using microbial cultures

Isolates of phenotypically distinct species, *Shewanella loihica*[45], *Roseobacter denitrificans*[46], *Pseudomonas stutzeri*[47], *Ruegeria pomeroyi*[48] and *Vibrio alginolyticus*[49], were acquired from the National Center for Marine Algae and Microbiota. Isolates of *Rhodoluna lacicola*[50] and *Oligotropha carboxidovorans*[51] were obtained from the German Collection of Microorganisms and Cell Cultures. Cultures were grown in specific media at varying temperatures and nutrient concentrations (medium recipes and growth conditions are provided in Supplementary Table 1) in 70 ml glass serum vials sealed with butyl rubber stoppers with atmospheric air in the headspace (starting O$_2$ concentration 21%) on a rotating shaker table to facilitate complete gas exchange between the culture medium and vessel headspace. A headspace approach was used owing to the need to withdraw small volumes of fluids at discrete timepoints.

The oxygen concentration in all culture vessels was measured using the Firesting-O$_2$ optical meter according to the manufacturer's instructions (PyroScience sensor technology). For each serum bottle, the optode probe was inserted into the headspace with the temperature-normalizing probe also inserted into a different bottle headspace (with seawater in the liquid phase) kept at the same temperature as the culture undergoing analysis. The oxygen concentration was measured in the headspace and converted to dissolved oxygen concentration in the culture medium, assuming complete gas equilibration between the culture medium and the culture headspace (as all cultures were kept on a rotating shaker table) and taking into account temperature, headspace pressure and salinity of the culture medium[52]. This conversion was possible due to the Firesting-O$_2$ optode method of directly measuring O$_2$ at the tip of the optode probe (that is, there is no diffusion or consumption of O$_2$ via this method). After measuring O$_2$ in the headspace, the concentration present in the culture medium was calculated using the headspace equilibration technique and calculated[53] using Bunsen coefficients[54,55] for O$_2$, which takes into account temperature and salinity at constant atmospheric pressure at sea level. Concentrations of O$_2$ in the culture medium were measured every few hours on the basis of the estimated doubling time of each culture type. Measurements of O$_2$ concentration for each culture spanned initial inoculation through the log phase of growth until the stationary phase was reached. Operational changes in microbial cell abundance were determined on the basis of measurements of optical density at a wavelength of 600 nm using a spectrophotometer. A culture was determined to be in the stationary phase once the O$_2$ concentration no longer decreased exponentially, and when the cell abundance stopped increasing exponentially. During the stationary phase, the rates of O$_2$ respiration were calculated as time-normalized differences in O$_2$ concentration between measurements. To estimate the rates of O$_2$ respiration per average cell, the abundance of microbial cells was determined by flow cytometry analysis of subsamples of the culture material taken at the same time as the O$_2$ concentrations were measured, as described below.

After the cultures reached stationary phase, 1 ml subsamples were withdrawn using a sterile needle and syringe, transferred to a 2 ml cryovial and amended with 1 μl RSG stock solution. After a gentle mix, the cryovials were incubated for 30 min at the same temperature as their source cultures. The cryovials were then amended with 5% glycerol and 1× pH 8 Tris-EDTA buffer (final concentrations), flash-frozen in liquid nitrogen and stored at −80 °C until downstream use in flow cytometry analysis. After RSG-incubated samples were taken and while the culture was still in the stationary phase, killed controls for each culture type were prepared by autoclaving 10 ml of culture and medium for 30 min at 121 °C. Samples of uninoculated medium and killed controls were also taken for use in determining appropriate gates for downstream flow cytometry analysis.

After thawing, the cryopreserved samples were analysed using the ZE5 Cell Analyzer flow cytometer (Bio-Rad). The samples were analysed with blue excitation (488 nm), and the instrument was triggered on both forward scatter and green fluorescence using a 525/35 nm bandpass filter. The analysis gate was defined on the basis of particle green (RSG) fluorescence and right-angle side scatter. The analysis gates were set to eliminate areas that contained only non-cellular particle noise from uninoculated media blanks, and also by eliminating areas that contained dead cells (killed by autoclaving). For all culture samples, an area of growing cells was clearly defined (Extended Data Fig. 2b). Cell abundance and green fluorescence were analysed in FlowJo v.10.6.2 (Becton Dickinson).

### RSG fluorescence normalization

To account for day-to-day drift and differences between the ZE5 Cell Analyzer flow cytometer (Bio-Rad) and the BD Influx Mariner flow cytometer (Becton Dickinson, formerly Cytopeia) cytometry instruments and settings, a procedure was developed to normalize cell green fluorescence using one of two fluorescent particle size standard kits: NFPPS-52-4K or RCP-30-5A (Spherotech). These beads were analysed using flow cytometry each day when RSG-labelled cells were analysed, using the same instrument and settings. For each bead, the geometric mean of the 525/35 fluorescence was calculated using FlowJo. For the NFPPS-52-4K bead kit, the least-fluorescent bead was assigned a normalized fluorescence value equal to 1 and the normalized fluorescence values of the other beads (Supplementary Table 5) were calculated as the average ratio of their fluorescence relative to the fluorescence of the least-fluorescent bead. Averages of these ratios were estimated from the application of this technique on multiple days.

For each day of analysis of field and culture samples, a linear regression was performed to establish the relationship between the measured bead fluorescence and their normalized fluorescence values. Then the normalized fluorescence of each cell was estimated as follows:

$$\text{Normalized fluorescence} = m \times x + b$$

where $x$ is a cell's measured fluorescence, $m$ is the experimentally determined slope of the bead linear regression from the day of cell analysis and $b$ is the experimentally determined intercept of the bead linear regression from the day of cell analysis.

To intercalibrate the two bead sets, they were flow cytometrically co-analysed on multiple days on the BD Influx Mariner flow cytometer and the normalized fluorescence of the geometric mean of each RCP-30-5A bead was calculated using the linear regression equation described above. A standardized normalized fluorescence value was calculated by averaging the geometric mean of RCP-30-5A beads from multiple days (Supplementary Table 5 (column RCP-30-5A)). The standardized normalized fluorescence of the RCP-30-5A beads was used to perform a linear regression analysis to establish the relationship between the measured bead fluorescence and their normalized fluorescence (see above) on days on which the NFPPS-52-4K beads were not available.

Subsequently, we correlated the normalized fluorescence of microbial cultures against their average respiration rate (Fig. 1a), resulting in the following exponential relationship:

$$\text{Cell respiration} = 0.004 \times e^{0.194 \times \text{normalized fluorescence}}$$

where the cell respiration is the cell-specific respiration estimate in fmol $O_2$ per cell per h and the normalized fluorescence is the cell's normalized RSG fluorescence (see above). This equation was transformed for ease of interpretation in Fig. 1a.

### Validation of RSG-based prokaryoplankton respiration rate measurements

To validate the use of RSG in quantifying respiration, we performed simultaneous RSG probing and bulk respiration measurements using traditional Winkler methodology[14] on geographically diverse, coastal and open ocean prokaryoplankton samples under in situ conditions (Supplementary Table 2). For GoM samples, seawater was collected at a depth of 1 m using a Niskin bottle at the same location as the RSG samples described above, except for one sample (Damariscotta River), the location of which is given in Supplementary Table 2. The seawater was collected, passed through a 40-µm-mesh filter and used to fill eight biological oxygen demand (BOD) bottles without headspace. Each flask was overflown by at least 3 times its volume to avoid air bubbles and to omit any additional gas exchange the water would have had with the atmosphere during transfer into the vessel. Four of these bottles were immediately fixed[56], while the remaining four bottles were incubated at in situ temperature in the dark for 24 h before fixation[56]. The oxygen concentration in all of the bottles was analysed using an amperometric titrator[56].

Open ocean samples were collected directly from the Niskin bottles into BOD flasks, whereas coastal Mediterranean Sea samples were collected with buckets from the shore and transferred to acid-cleaned polycarbonate carboys. Once at the laboratory, the coastal Mediterranean Sea water was filtered through 0.8 µm polycarbonate filters and transferred to BOD flasks. Each flask was overflown by at least three times its volume. The three replicate BOD flasks were fixed immediately, and another three flasks were incubated in the dark at in situ temperature for 24 h. After fixation with the Winkler chemicals, the samples were measured using the potentiometric method[56] (coastal Mediterranean and North Atlantic) or spectrophotometry[57] (south and equatorial Atlantic). Winkler-based prokaryoplankton respiration rates were estimated as the difference in dissolved oxygen concentration between initial and final samples.

Immediately after collecting the field sample, subsamples were incubated with RSG at in situ temperature in the dark for 30 min, then stored at −80 °C until flow cytometry analyses and estimates of cell-specific respiration rates as described above. The RSG-based bulk respiration rates were obtained by multiplying the average, cell-specific respiration rates (described above) by the abundance of respiring cells in each sample.

### Single-cell genome and cell diameter analyses of marine prokaryoplankton

For the main study, samples from the GoM were collected from the dock of Bigelow Laboratory for Ocean Sciences on the shore of East Boothbay at the mouth of the Damariscotta River (43.8609° N, 69.5782° W) on six days from April 2017 to July 2019 (Supplementary Table 2). Water samples were collected from a depth of 1 m with a 5 l Niskin bottle, passed through a 50-µm-mesh filter, transferred to acid-washed polycarbonate bottles, transported to the lab at in situ temperature and processed immediately. Open ocean samples were collected with 12 l Niskin bottles attached to a CTD rosette and transferred to acid-washed polycarbonate bottles. Subsequently, duplicate 1 ml seawater subsamples were transferred to cryovials, amended with 1 µl RSG (1 µM final concentration), incubated in the dark at in situ temperature for 30 min, amended with 5% glycerol and 1× pH 8 TRIS-EDTA buffer (final concentrations), flash-frozen in liquid $N_2$ and stored at −80 °C. In all cases, additional 1 ml seawater subsamples were amended with 5% glycerol and 1× pH 8 TRIS-EDTA buffer (final concentrations), flash frozen in liquid $N_2$ and stored at −80 °C without RSG addition.

To compare day and night conditions, additional GoM seawater samples were collected from the Bigelow Laboratory dock according to the same procedures as described above on 24 (14:00 EST) and 25 (02:00 EST) August 2021. Triplicate 1 ml sample aliquots were labelled with RSG and cryopreserved as described above.

For sorting of respiring cells, we used samples that were labelled with RSG in the field, as described above. For non-specific sorting of prokaryoplankton cells, the cryopreserved and thawed seawater samples were incubated with the SYTO 9 nucleic acid stain (5 µM final concentration; Thermo Fisher Scientific) on ice for 10–60 min. Flow cytometry analysis and sorting was performed using a BD InFlux Mariner flow cytometer equipped with a 488 nm laser for excitation and a 70 µm nozzle orifice (Becton Dickinson, formerly Cytopeia). The cytometer was triggered on forward scatter in most cases and on green fluorescence in August 2021. The 'single-1 drop' mode was used for maximal sort purity. The sort gate was defined based on particle green fluorescence (proxy to either nucleic acid content or respiration rate), forward scatter (proxy to cell diameter) and the ratio of green versus red fluorescence (for improved discrimination of cells from detrital particles). Cells were deposited into 384-well microplates containing 600 nl per well of 1× TE buffer and stored at −80 °C until further processing. Of the 384 wells, 317 wells were dedicated for single cells, 64 wells were used as negative controls (no droplet deposition) and 3 wells received 10 cells each to serve as positive controls. The accuracy of droplet deposition into microplate wells was confirmed several times during each sort day by sorting 3.46 µm diameter SPHERO Rainbow Fluorescent Particles (Spherotech) and examining their presence at the bottom of each well using microscopy. In these examinations, fewer than 2% of wells did not contain beads and <0.4% wells contained more than one bead.

Index sort data were collected using the BD Sortware software. The following laboratory cultures were used in the development of a cell diameter equivalent calibration curve: *Prochlorococcus marinus* CCMP 2389; *Microbacterium* sp.; *Pelagibacter ubique* HTCC1062; and *Synechococcus* CCMP 2515. The average cell diameters of these cultures were determined using the Multisizer 4e Coulter Counter (Beckman Coulter). The average forward scatter of each of the four cultures was determined using the same flow cytometry settings used during environmental sample sorting, repeated on each day of single-cell sorting. We observed a strong correlation between cell diameters and forward scatter (FSC) among these cultures[16]. Taking advantage of this correlation, the estimated diameter of the sorted environmental cells (diameter, in µm) was estimated from a log-linear regression model:

$$\text{diameter} = 10^{(a \times \log_{10}[\text{FSC}] - b)},$$

where $a$ and $b$ are empirically derived regression coefficients from each day of sorting[16], and FSC is the measured forward scatter. This calculation was repeated for all flow cytometry cell sorting sessions. The proxy cell volume was calculated assuming a spherical shape for all lineages.

Before genomic DNA amplification, cells were lysed and their DNA was denatured by two freeze–thaw cycles and the addition of 700 nl of a lysis buffer consisting of 0.4 M KOH, 10 mM EDTA and 100 mM dithiothreitol, and a subsequent 10 min incubation at 20 °C. The lysis was terminated by the addition of 700 nl of 1 M Tris-HCl, pH 4. Single-cell whole-genome amplification was performed using WGA-X[16]. In brief, the 10 µl WGA-X reactions contained final concentrations of 0.2 U µl$^{-1}$ Equiphi29 polymerase (Thermo Fisher Scientific), 1× Equiphi29 reaction buffer (Thermo Fisher Scientific), 0.4 µM each dNTP (New England BioLabs), 10 µM dithiothreitol (Thermo Fisher Scientific), 40 µM random heptamers with two 3′-terminal phosphorothioated nucleotide bonds (Integrated DNA Technologies) and 1 µM SYTO 9 (Thermo Fisher Scientific). These reactions were performed at 45 °C for 12–16 h, then inactivated by a 15 min incubation at 75 °C. To prevent WGA-X reactions from being contaminated with non-target DNA, all cell lysis and DNA amplification reagents were treated with ultraviolet light in a Stratalinker (Stratagene)[58] system. An empirical optimization of the ultraviolet exposure was performed to determine the length of ultraviolet exposure that is necessary to cross-link all detectable contaminants without inactivating the reaction. Cell sorting, lysis and WGA-X set-up were performed in a HEPA-filtered environment conforming to Class 1000 cleanroom specifications. Before cell sorting, the instrument, the reagents and the workspace were decontaminated for DNA using ultraviolet irradiation and sodium hypochlorite solution[59]. To further reduce the risk of DNA contamination, and to improve accuracy and throughput, Bravo (Agilent Technologies) and Freedom Evo (Tecan) robotic liquid handlers were used for all liquid handling in 384-well plates.

Libraries for SAG genomic sequencing were created with Nextera XT (Illumina) reagents according to the manufacturer's instructions except for the purification steps, which were performed using column clean-up kits (QIAGEN) and library size selection, which was performed with BluePippin (Sage Science) with a target size of 500 ± 50 bp. DNA concentration measurements were performed using Quant-iT dsDNA Assay Kits (Thermo Fisher Scientific) according to the manufacturer's instructions. Libraries were sequenced with either the NextSeq 500 (Illumina) system in 2 × 150 bp mode or the NextSeq 2000 (Illumina) system in 2 × 100 bp mode. The obtained sequence reads were quality-trimmed using Trimmomatic (v.0.32)[60] using the following settings: -phred33 LEADING:0 TRAILING: 5 SLIDINGWINDOW:4:15 MINLEN:36. Reads matching the *Homo sapiens* reference assembly GRCh38 and a local database of WGA-X reagent contaminants[16] (≥95% identity of ≥100 bp alignments) as well as low-complexity reads (containing <5% of any nucleotide) were removed. The remaining reads were digitally normalized using kmernorm v.1.05 (http://sourceforge.net/projects/kmernorm) using the settings -k 21 -t 30 -c 3 and then assembled with SPAdes (v.3.0.0)[61] using the following settings: --careful --sc --phred-offset 33. Each end of the obtained contigs was trimmed by 100 bp and only contigs longer than 2,000 bp were retained. Contigs matching the *H. sapiens* reference assembly GRCh38 and a local database of WGA-X reagent contaminants[16] (≥95% identity of ≥100 bp alignments) were removed. The quality of the resulting genome assemblies was determined using CheckM (v.1.0.7)[62] and tetramer frequency analysis[63]. This workflow was evaluated for assembly errors using three bacterial benchmark cultures with diverse genome complexity and percentage GC, indicating no non-target and undefined bases in the assemblies and the following average frequencies of misassemblies,

indels and mismatches per 100 kb: 1.5, 3.0 and 5.0, respectively (ref. [16]). The 153 SAGs in which potentially contaminating DNA sequences were detected were removed from the dataset. This resulted in 7,518 curated SAG genome assemblies.

Pairwise SAG average amino acid identity was estimated using CompareM[64]. The 16S rRNA gene regions longer than 500 bp were identified using local alignments provided by BLAST against CREST's[65] curated SILVA reference database SILVAMod (v.128)[66] and classified using a reimplementation of CREST's last common ancestor algorithm. Phylogenetic trees were generated from near-complete (>1,400 bp) SAG 16S rRNA genes by first producing alignments using the SINA aligner (v.1.2.11)[67] and then inferring maximum-likelihood phylogenetic relationships in MEGACC (v.10.2.4)[68] with 100 bootstraps. The trees were annotated and visualized in iTOL[69]. Taxonomic assignments of SAGs were obtained with GTDB-Tk (v.1.4.1)[70].

Functional annotation was first performed using Prokka[71] with the default Swiss-Prot databases supplied by the software. Prokka was run a second time with a custom protein annotation database built from compiling Swiss-Prot[72] entries for Archaea and Bacteria. Taxonomic assignments were obtained using GTDB-Tk (v.1.4.1)[70]. Viral sequences (whole or partial contigs) were independently identified using three pre-existing bioinformatic viral identification tools: ViruScope[73] (virus_prob > 0.95), VirSorter2[74] (category 1 and 2 only) and DeepVirFinder[75] ($P > 0.95$). The results of these three tools were combined. False-positives ($P < 0.95$) were removed using CheckV[76].

## Metagenomics and metatranscriptomics

Microbial biomass from 100–200 ml seawater aliquots was collected on Supor 0.2 µm sterilized filters (Pall Corporation) by vacuum filtration (max pressure of 15 psi), then flash-frozen in liquid nitrogen and stored at −80 °C until nucleic acid extraction. MTST5 and MTST6 RNA standards were added to each filter before RNA extraction to an estimated concentration of 0.1–1% total yield, as previously described[31]. DNA and RNA were extracted in parallel using the Zymobiomics DNA/RNA Miniprep kit (Zymo Research) and stored at −80 °C until further processing. RNA samples were further cleaned and concentrated using the RNA Clean & Concentrator kit (Zymo Research). cDNA libraries for Illumina shotgun sequencing were generated with the KAPA RNA HyperPrep (Roche Sequencing). Libraries for DNA shotgun sequencing were generated using the Nextera XT kit (Illumina) according to the manufacturer's instructions. Both library types were size-selected with BluePippin (Sage Science) set for 370 bp 'tight' and quantified using the Tapestation (Agilent). These libraries were sequenced using the NextSeq 500 (Illumina) system in 2 × 150 bp mode.

After sequencing, reads from both RNA and DNA libraries were initially trimmed with Trimmomatic (settings: -phred33 LEADING:0 TRAILING:5 SLIDINGWINDOW:4:15 MINLEN:75). Reads matching the *H. sapiens* reference assembly GRCh38 and a local database of WGA-X reagent contaminants[16] (≥95% identity of ≥100 bp alignments), as well as low-complexity reads (containing <5% of any nucleotide) were removed. The remaining reads were merged with bbmerge using the following settings: k = 40, extend2 = 60, iterations = 5, loose='t', qtrim2='t'. The merged reads were taxonomically and functionally annotated using the Kaiju classifier[77], after replacing the underlying reference genome database with SAGs from this study as previously described[3]. To identify SSU rRNA gene transcripts, metatranscriptomic reads were mapped on the SILVA SSU library[66] (v.132) using Bowtie2[78] at 95% identity across the length of the read. The abundance of each genus in a sample ($GA_i$; cells per ml) was calculated as follows:

$$GA_i = \left[ \left( \left( \frac{a_i}{b_i} \right) \frac{1}{c} \right) \frac{1}{d_i} \right] \frac{f}{e}$$

where $e$ is a normalization factor for genera that did not generate SAGs, and is defined as:

$$e = \sum_{i=1}^{N}\left[\left(\left(\frac{a_i}{b_i}\right)\frac{1}{c}\right)\frac{1}{d_i}\right],$$

where $a_i$ is the count of metagenome reads recruited to genus $i$ SAG, $b_i$ is the average fraction of nucleotides in genus $i$ SAG assemblies that encode proteins, $c$ is the total count of reads in the metagenome, $d_i$ is the average estimated genome size of the genus $i$ SAG and $f$ is the total abundance of prokaryoplankton, in cells per ml. This calculation was repeated for every sampling day.

The average count of transcripts per cell (TC$_i$) was calculated for each genus and gene as follows:

$$TC_i = a_i \times \left(\frac{b}{c}\right) \times \left(\frac{1}{d}\right) \times \left(\frac{1}{GA_i}\right),$$

where $a_i$ is the count of reads mapped on the gene of interest, $b$ is the number of MTST5 and MTST6 RNA transcripts added to the sample, $c$ is the number of reads mapped on MTST5 and MTST6 and so the ratio $b/c$ is the number of transcripts per read, $d$ is volume of sample collected on the filter in ml and GA$_i$ is the genus abundance (cells per ml; see above). This calculation was repeated for every sampling day.

## Genus-specific respiration estimates

The average cellular respiration rate (ACR$_i$; amol O$_2$ per cell per h) of each genus in each GoM seawater sample was estimated as follows:

$$ACR_i = \left(\frac{a_i R + 0.5 \times z[GA_i - R]}{GA_i}\right)$$

where $R$ is defined as:

$$R = b_i/c \times d,$$

where $a_i$ is the average respiration rate of a given genus' cells recovered in SAGs with estimated respiration rates exceeding the detection limit (4 amol O$_2$ per cell per h), $b_i$ is the count of SAGs classified as belonging to genus $i$ (that is, the number of active cells in genus $i$), $c$ is the count of all RSG SAGs (also known as the number of sequenced active cells in sample), so $b_i/c$ is the fraction of active cells in genus $i$, $d$ is the total concentration of active cells with an estimated respiration rate above the detection limit (cells per ml; Supplementary Table 2) and $z$ is equal to the minimum detected cell-specific respiration rate, in amol O$_2$ per cell per h.

The quantity $R$ is the number of active cells of genus $i$ per ml. The quantity $0.5 \times z[GA_i - R]$ is a correction factor for the cells below the detection limit. Owing to different sensitivities of the measurement methods, it was possible for $R$ to occasionally be slightly greater than GA$_i$. When this happened, $R$ was set equal to GA$_i$ making GA$_i - R = 0$. This calculation was repeated for every sampling day. This calculation was only made for those genera and samples for which at least 3 RSG-labelled SAGs were identified.

## Water chemistry analyses

For major nutrient concentration measurements, 100 ml subsamples of GoM seawater were collected from the same samples that were used in single-cell genomics and respiration analyses. These samples were immediately filtered through a nucleopore track-edge membrane 25 mm, 0.4 μm pore-size filter (Whatman) and stored at −80 °C until analysis of NO$_3^-$, NO$_2^-$, NH$_4^+$ and PO$_4^{3-}$ on a SEAL AA3 (Seal Analytical Limited). Moreover, triplicate seawater subsamples, 100 ml each, were collected from each field sample and vacuum-filtered onto glass microfiber filters (Whatman, GF/F) within 2 h. Each GF/F filter was extracted in 10 ml of 90% acetone at −20 °C for 24–48 h. The extracts were analysed

on the 10-AU fluorometer (Turner Designs) before and after the addition of 50 μl of 10% HCl, and the concentrations of chlorophyll and phaeophytin were determined as described previously[79]

## Reporting summary

Further information on research design is available in the Nature Portfolio Reporting Summary linked to this article.

## Data availability

All metagenome and metatranscriptome reads and single-cell genome assemblies >100 kb are available under NCBI BioProject PRJNA846736. All metagenome and metatranscriptome reads and single-cell genome assemblies, including the 603 genome assemblies <100 kb are available at the Open Science Framework (https://osf.io/r2un6).

## Code availability

All analyses were performed using open-access software. Custom scripts used to combine results and generate images are available on request.

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

**Acknowledgements** We thank B. Thompson, C. Mascena and B. Tupper at the Single Cell Genomics Center of the Bigelow Laboratory for Ocean Sciences for the generation of single-cell genomic data; M. A. Moran, C. Smith and B. Nowinski for their help establishing a quantitative transcriptomics workflow; M. A. Moran and S. Giovannoni for their comments on the manuscript; the captains and crews of R/V *Sarmiento de Gamboa*, R/V *Ramon Margalef* and R/V *Atlantis* for their support at sea; K. Ziervogel, K. Dykens and D. Moser for their assistance in sample collection during AT42-11 aboard the R/V *Atlantis*; C. González-Pola and J. M. Arrieta for offering the opportunity to participate in their cruises. The INOCEN laboratory (IEO, PI M. Álvarez) accounted for the oxygen analysis in RADPROF 2018. R. Santiago and A. Massanet for their help with chemical analysis of Mallorca samples. Seawater samples from the GoM were collected off the dock of the Bigelow Laboratory in East Boothbay and did not require a permit. Seawater samples from AT42-11 were collected with the consent of the Government of Canada. This work was supported by awards from the US National Science Foundation (1826734 to R.S., D.E., B.N.O. and N.J.P.; 1737017 to B.N.O.; and 1335810 to R.S.), the Austrian Science Fund (FWF) project ARTEMIS (P28781-B21 to G.J.H.) and by the Simons Foundation grant (827839 to R.S.).

**Author contributions** R.S., B.N.O., N.J.P., D.E. and G.J.H. conceived the study and secured funding. M.R.L., T.D. and E.S. maintained cultures and measured respiration. J.H.M.-M., M.R.L., E.S., T.D., L.C.L., N.J.P. and R.S. collected the samples and generated the data. J.H.M.-M., M.R.L., J.M.B., T.D., J.B. and P.T. analysed the data. J.H.M.-M., M.R.L., N.J.P. and R.S. wrote the paper with input from all of the authors.

**Competing interests** The authors declare no competing interests.

**Additional information**
**Correspondence and requests for materials** should be addressed to Ramunas Stepanauskas.

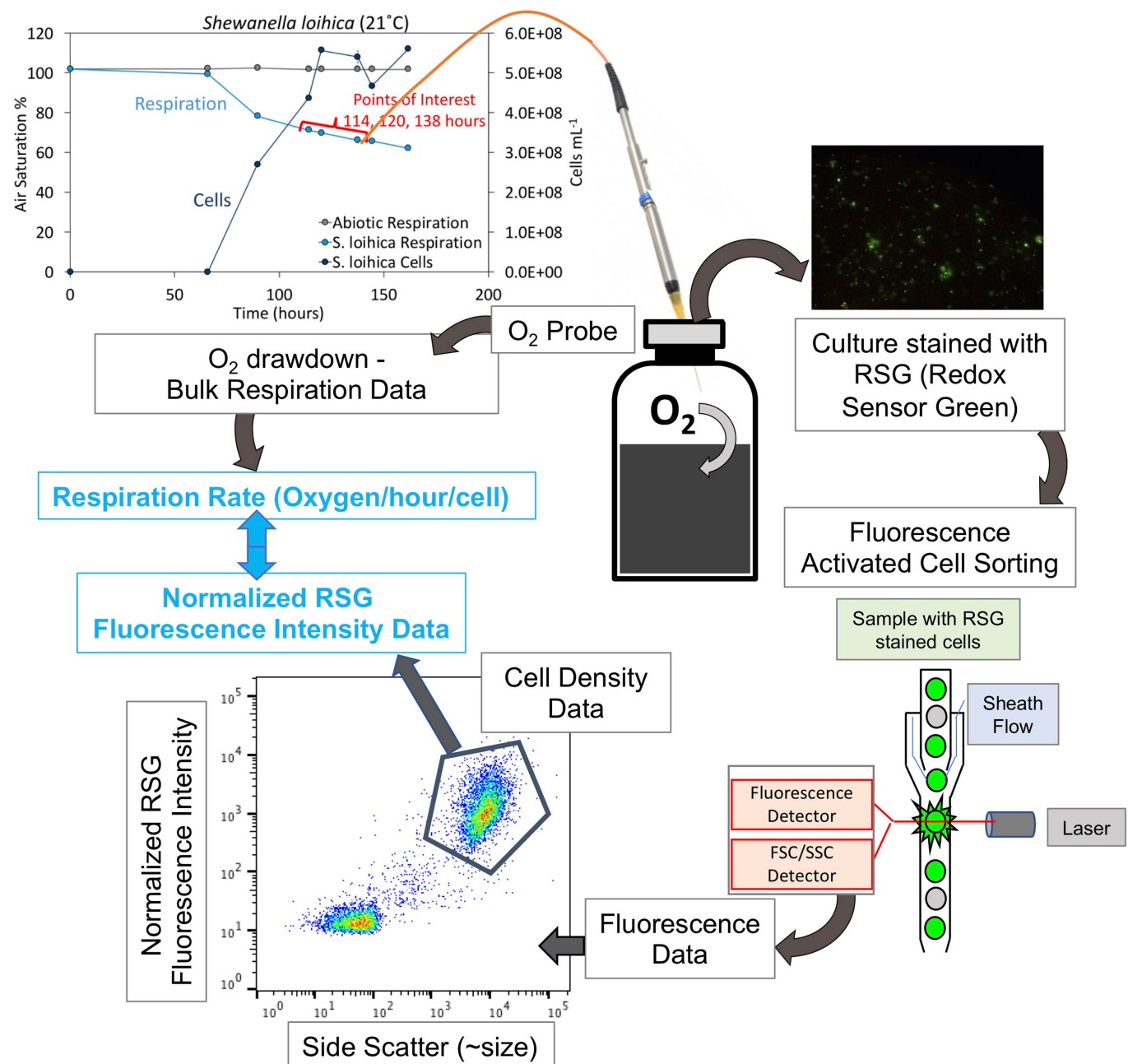

**Extended Data Fig. 1 | Schematic representation of the workflow to calibrate cell-specific respiration against RSG fluorescence.** Over a time course spanning inoculation to stationary phase, oxygen concentration was measured in sealed bottles containing cultured isolates of aquatic microorganisms. Subsamples of these incubations were probed with RSG, after which cell fluorescence and abundance were measured by flow cytometry. A calibration curve was developed by comparing the average cell fluorescence intensity against oxygen consumption that was normalized for cell abundance in each culture.

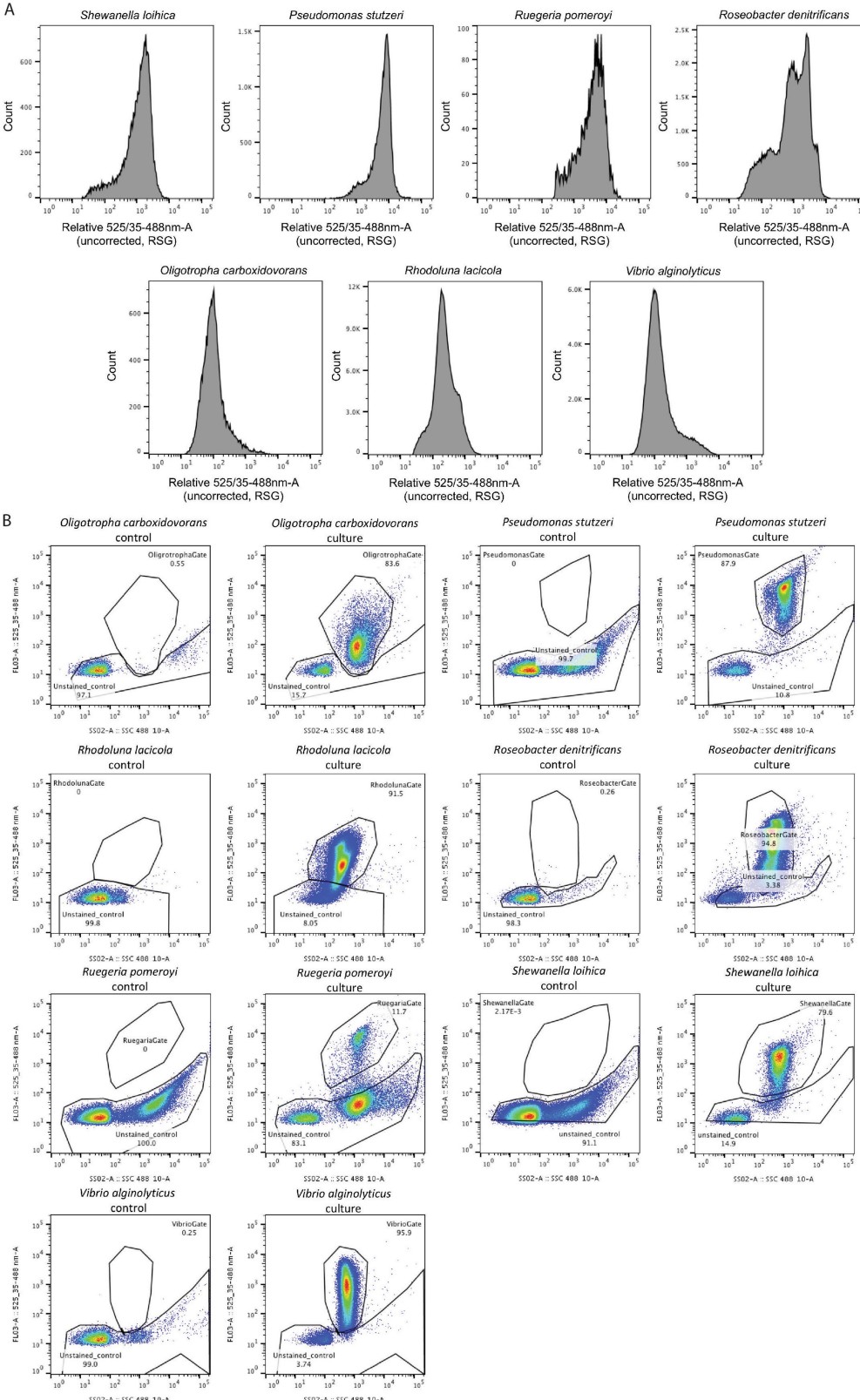

**Extended Data Fig. 2 | Flow-cytometric analyses of cultured isolates used in the respiration calibration. A**. Diversity of RSG fluorescence among cells in cultured isolate incubations. **B**. Setting gates of RSG-positive cells in scatterplots of light side scatter (x-axis) and green fluorescence (y-axis) of individual cells. Each culture was gated individually, based on a killed control sample of that culture.

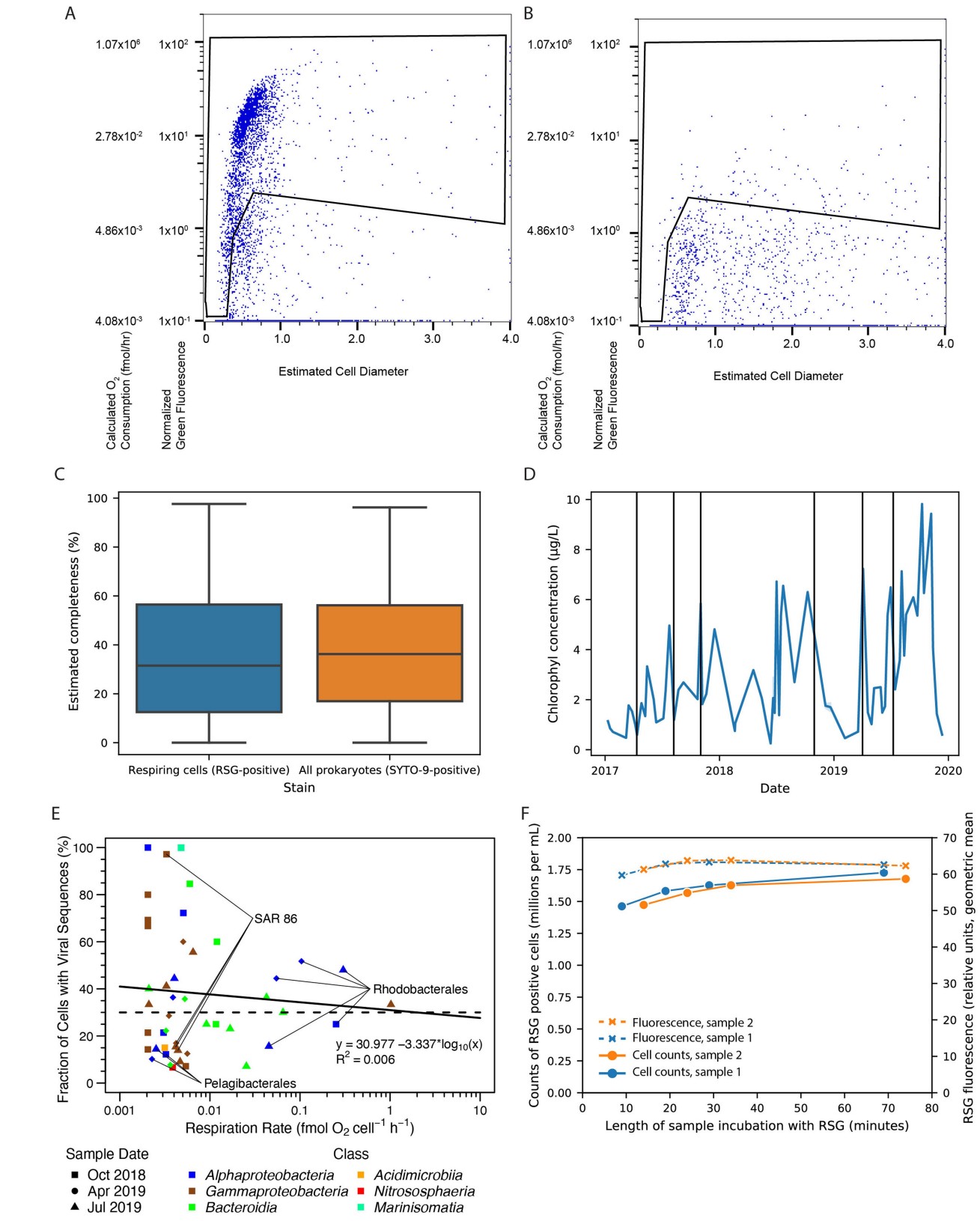

**Extended Data Fig. 3 | Contextual information. A-B**. Flow-cytometric gating of RSG-labelled prokaryoplankton. The GoM seawater sample was collected on April 2, 2019, and labelled with RSG before (**A**) or after (**B**) a heat treatment. Shown are the first 100,000 events in both samples (the majority of events are on the baseline). **C**. Genome assembly completeness of all Gulf of Maine SAGs that were probed with either RSG (blue, n = 2,572) or SYTO-9 (orange, n = 1,632). Boxes indicate means, first and third quartiles, and whiskers extending up to 1.5 inter-quartile ranges. **D**. Chlorophyl concentration near Boothbay Harbor (43.84° N, 69.64° W). Vertical black lines indicate days when samples for SAG analyses were collected. **E**. Relationship between respiration rates and the presence of virus DNA in individual cells. Each datapoint represents a genus in a single field sample. **F**. Optimization of RSG incubation length for marine prokaryoplankton. Indicated are the abundance and fluorescence intensity of RSG-positive cells in two replicate samples.

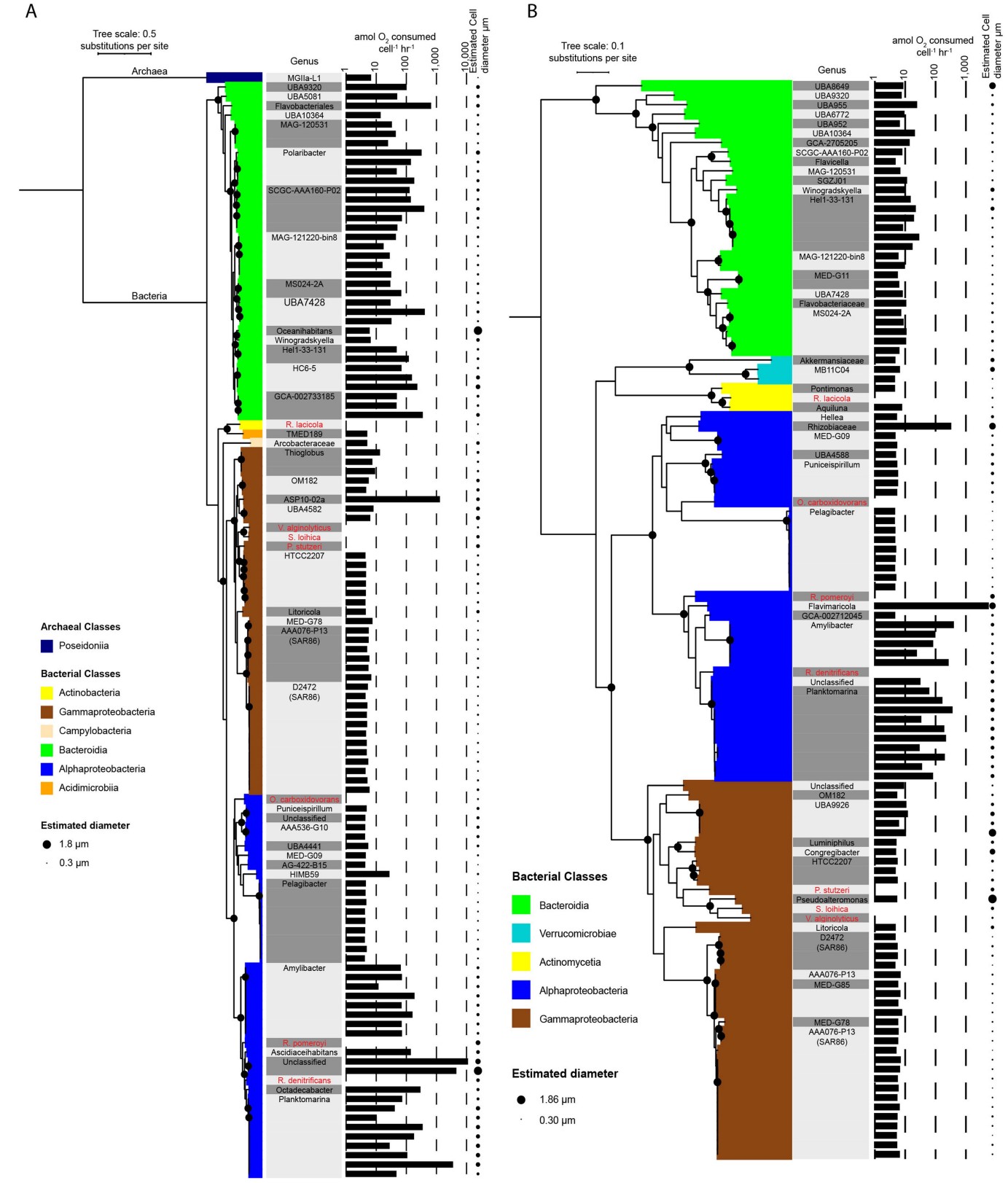

**Extended Data Fig. 4 | Maximum likelihood phylogenetic trees of the 16S rRNA genes from single cells obtained from the Gulf of Maine in 2019, complemented with cell respiration and diameter measurements. A.** SAGs from April 2, 2019. **B.** SAGs from July 9, 2019. In both trees, cultured isolates are included as phylogenetic references and are indicated by red font colour. The trees were rooted in the Archaeal branch, when present, and at the midpoint when no archaea are included.

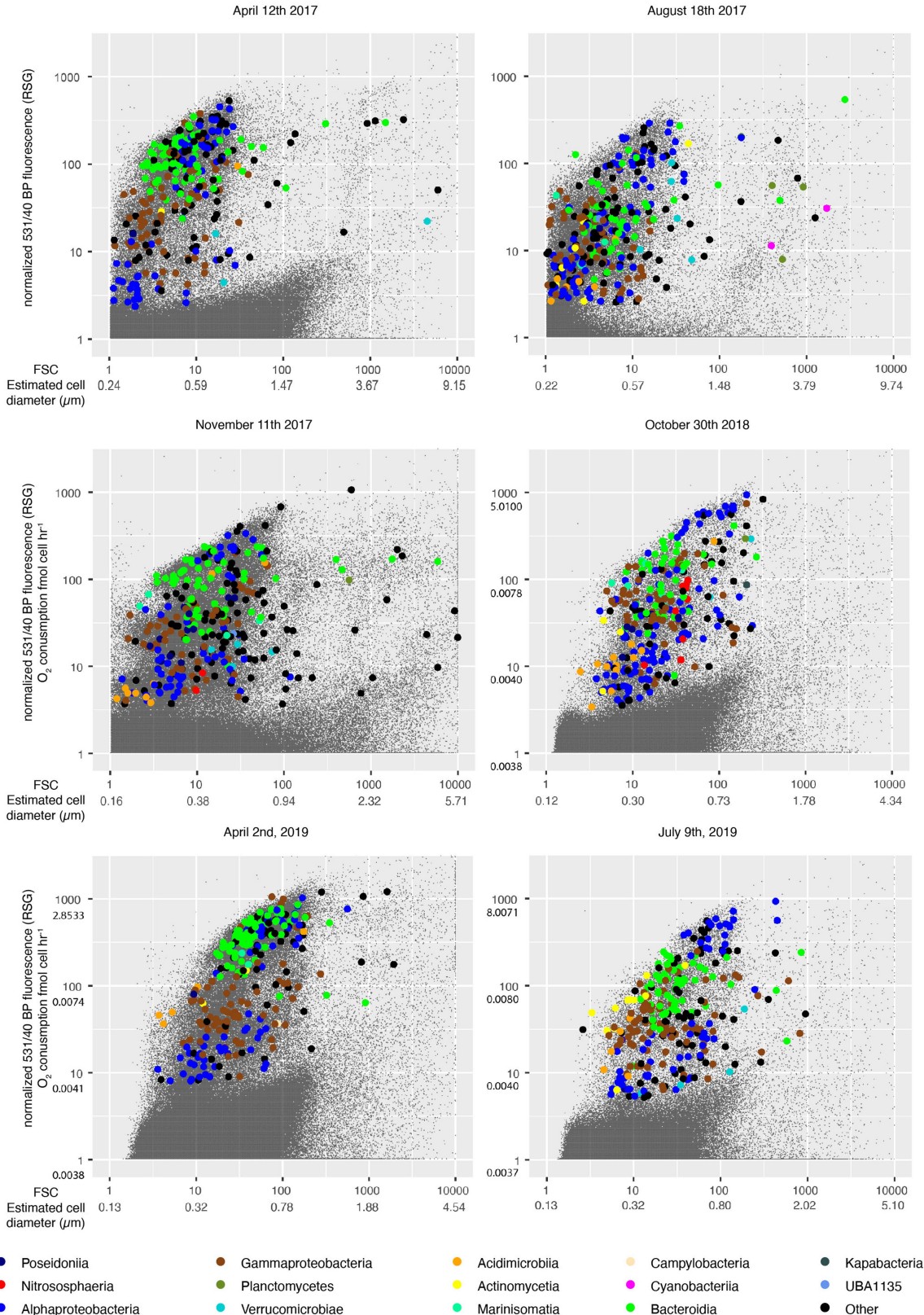

**Extended Data Fig. 5 | Estimates of diameters and respiration rates of individual cells of Gulf of Maine prokaryoplankton.** Cells are coloured by taxonomic class, which were determined by single cell genome sequencing.

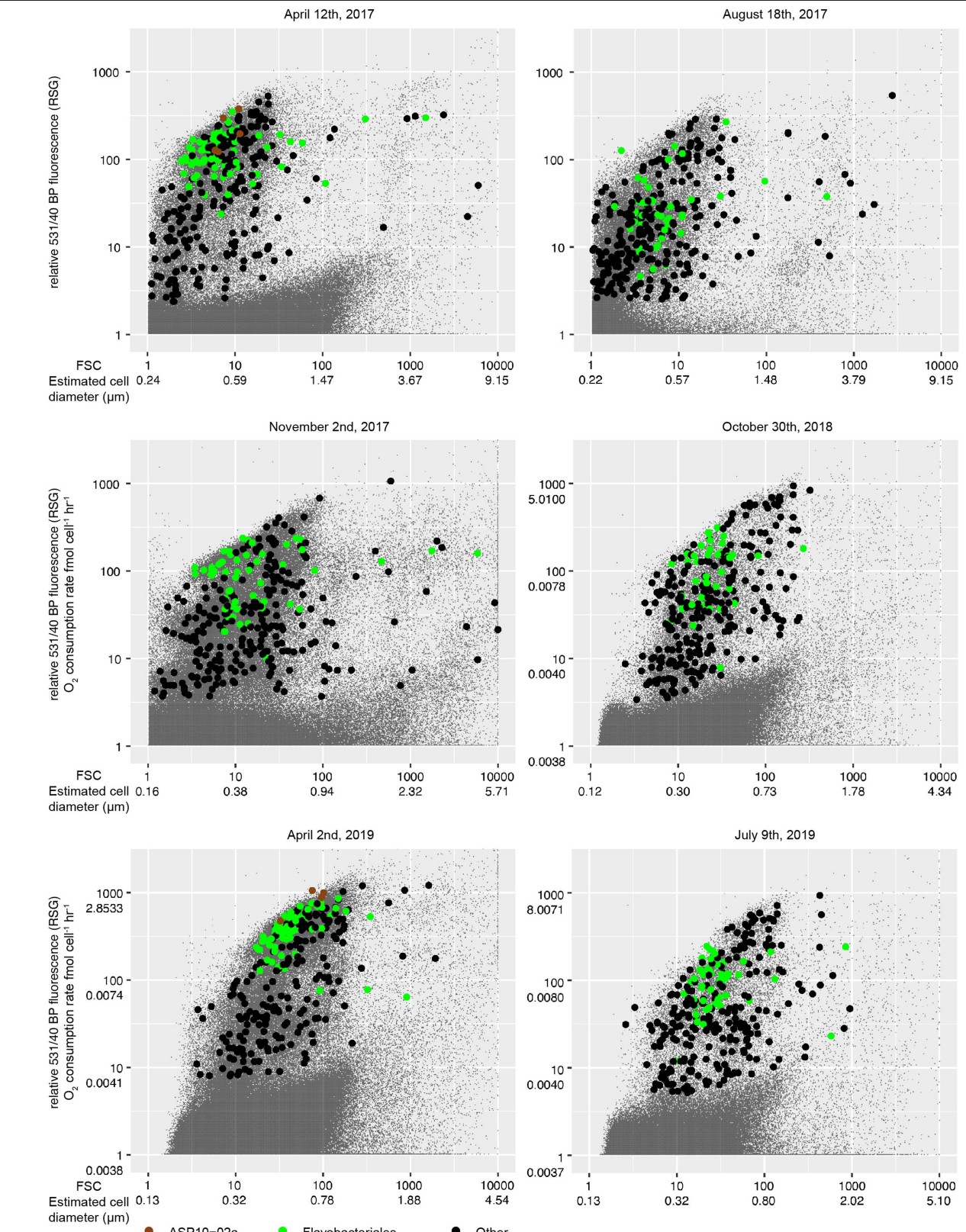

**Extended Data Fig. 6 | Estimates of diameters and respiration rates of individual cells of Gulf of Maine prokaryoplankton from genera with seasonally variable respiration rates.** Large dots represent cells that were sorted for single cell genomics. Highlighted in colour are lineages with pronounced differences in respiration rates among sampling dates. Due to changes in flow cytometer configuration, we could not estimate respiration rates in the three 2017 samples.

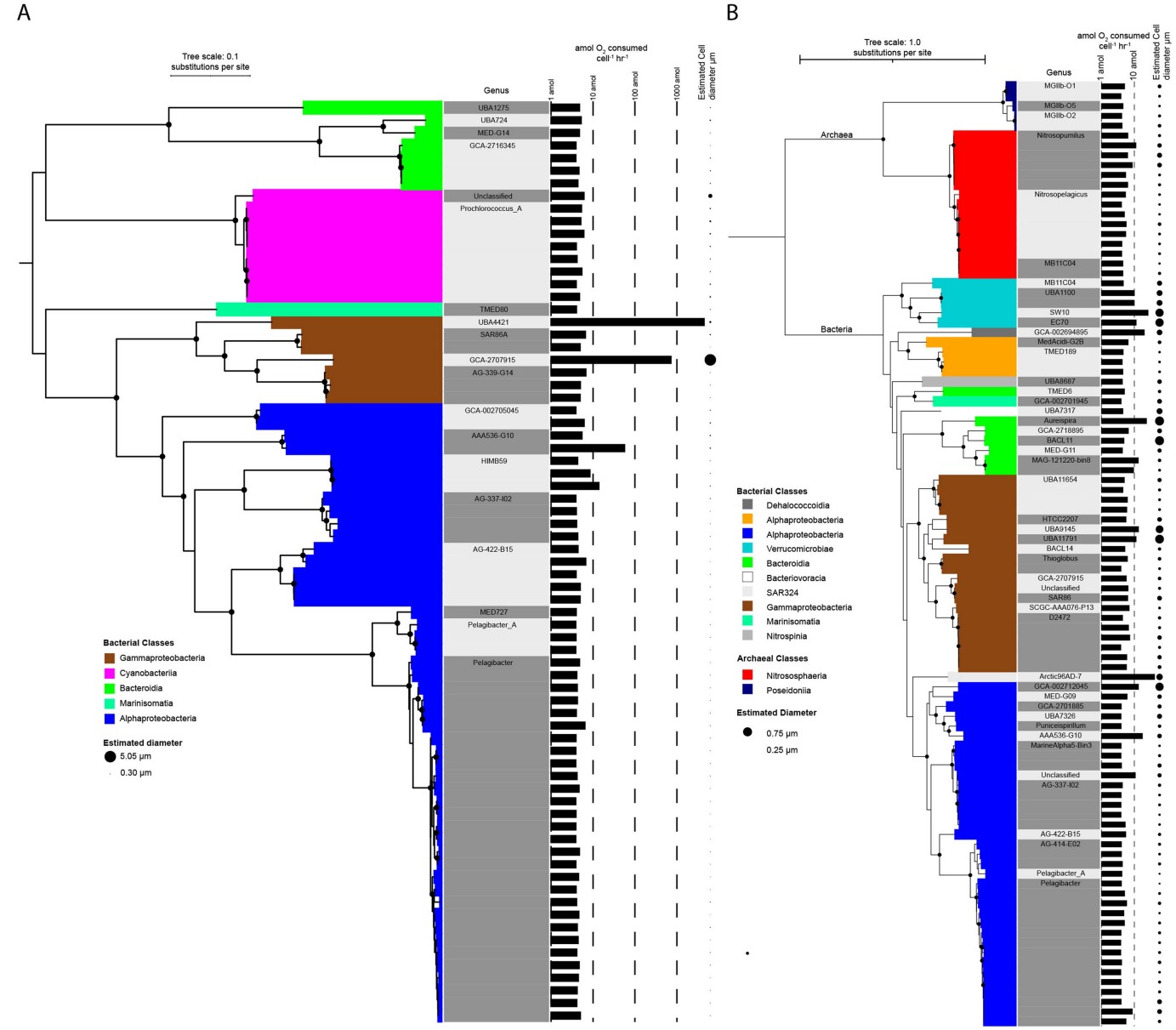

**Extended Data Fig. 7 | Maximum likelihood phylogenetic trees of the 16S rRNA genes from single cells of Atlantic prokaryoplankton, complemented with cell respiration and diameter measurements. A**. Cells from a sample collected in the southern Atlantic Ocean (32.35S, 44.02W) on March 9, 2019. **B**. Cells from a sample collected in the northern Atlantic Ocean (43.00N, 11.33W) on August 22, 2018. Trees were rooted at the midpoint. Bootstrap values >0.75 are indicated by black circles.

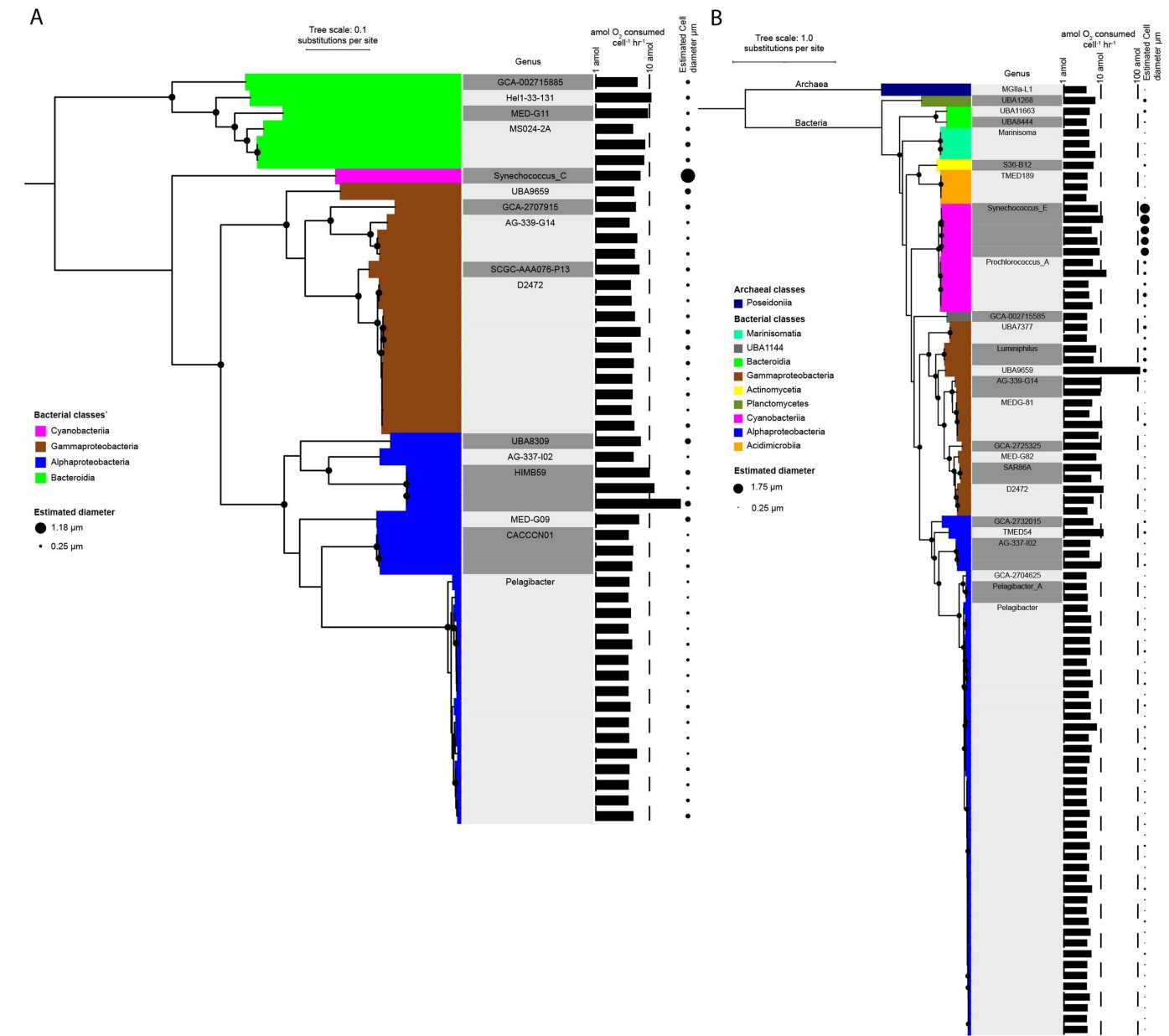

**Extended Data Fig. 8 | Maximum likelihood phylogenetic trees of the 16S rRNA genes from single cells of oceanic prokaryoplankton, complemented with cell respiration and diameter measurements. A.** Cells from a sample collected in the northeastern Pacific Ocean (47.76N, 127.76E) on May 23, 2019).

**B.** Cells from a sample collected in the equatorial Atlantic (3.10N, 29.01W) on March 20, 2019. Trees are rooted in the archaeal branch when present and at the midpoint when the archaeal branch is missing. Bootstrap values >0.75 are indicated by black circles.

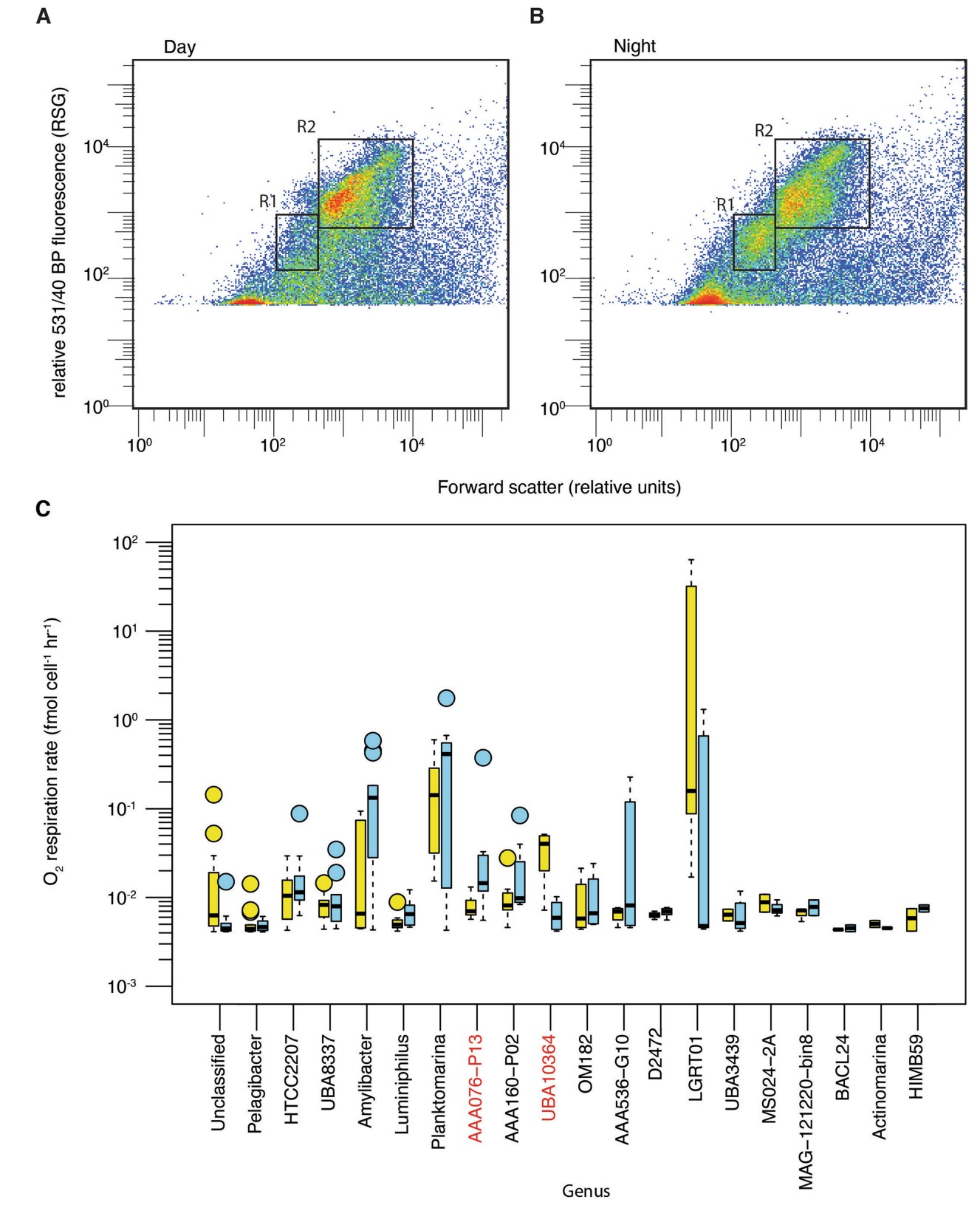

**Extended Data Fig. 9 | Comparison of the Gulf of Maine prokaryoplankton respiration rates during day and night. A**. Light forward scatter and RSG fluorescence of cells collected during day. **B**. Light forward scatter and RSG fluorescence of cells collected during night. Label "R1" indicates the position of cells with intermediate respiration rates that had higher abundance during night as compared to day. Label "R2" indicates positions of cells with high respiration rates. **C**. Genus-specific respiration rates during the day (yellow, n = 190) and night (blue, n = 190). Shown are genera from which at least three RSG-positive cells were sequenced. Red font indicates genera with a statistically significant difference in respiration rates between day and night (2-sided Mann-Whitney U-test, minimum 4 cells per time point, AAA076-P13 p-value = 0.010, UBA10364 p-value = 0.038). Boxes indicate the median, first and third quartile, and whiskers extending up to 1.5 times the inter quartile range (IQR). Values above or below 1.5 IQR are represented as circles.

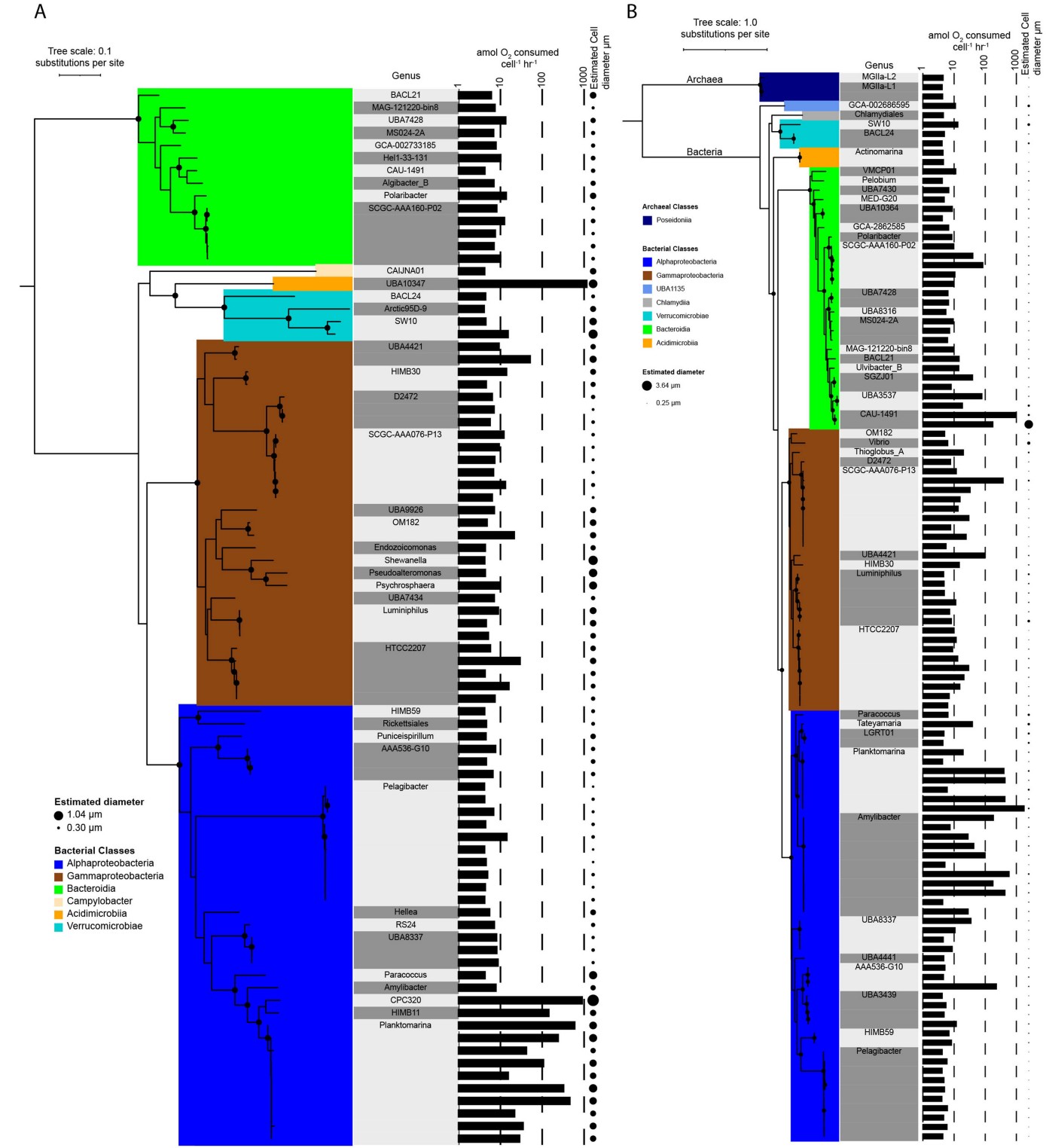

**Extended Data Fig. 10 | Maximum likelihood phylogenetic trees of the 16S rRNA genes from single cells collected in the Gulf of Maine in 2021 over a diurnal cycle, complemented with cell respiration and diameter measurements. A.** Cells from a sample collected at 14:00 on August 24, 2021.

**B.** Cells from a sample collected at 02:00 on August 25, 2021. Bootstrap values >0.75 are indicated by black circles. Trees are rooted in the archaeal branch when present and at the midpoint when the archaeal branch is missing.

# Reporting Summary

## Statistics

For all statistical analyses, confirm that the following items are present in the figure legend, table legend, main text, or Methods section.

| n/a | Confirmed | |
|---|---|---|
| ☐ | ☒ | The exact sample size (*n*) for each experimental group/condition, given as a discrete number and unit of measurement |
| ☐ | ☒ | A statement on whether measurements were taken from distinct samples or whether the same sample was measured repeatedly |
| ☐ | ☒ | The statistical test(s) used AND whether they are one- or two-sided <br> *Only common tests should be described solely by name; describe more complex techniques in the Methods section.* |
| ☐ | ☒ | A description of all covariates tested |
| ☐ | ☒ | A description of any assumptions or corrections, such as tests of normality and adjustment for multiple comparisons |
| ☐ | ☒ | A full description of the statistical parameters including central tendency (e.g. means) or other basic estimates (e.g. regression coefficient) AND variation (e.g. standard deviation) or associated estimates of uncertainty (e.g. confidence intervals) |
| ☐ | ☒ | For null hypothesis testing, the test statistic (e.g. *F*, *t*, *r*) with confidence intervals, effect sizes, degrees of freedom and *P* value noted <br> *Give P values as exact values whenever suitable.* |
| ☒ | ☐ | For Bayesian analysis, information on the choice of priors and Markov chain Monte Carlo settings |
| ☒ | ☐ | For hierarchical and complex designs, identification of the appropriate level for tests and full reporting of outcomes |
| ☒ | ☐ | Estimates of effect sizes (e.g. Cohen's *d*, Pearson's *r*), indicating how they were calculated |

*Our web collection on statistics for biologists contains articles on many of the points above.*

## Software and code

Policy information about availability of computer code

| Data collection | Data was generated using the commercially available programs FlowJo (BD; for Flow cytometry analyses), and NextSeq (Illumina; for DNA sequencing). |
|---|---|
| Data analysis | Data analysis was performed using multiple commercial and open source software (described in materials and methods) and custom scripts to process outputs. Custom code is available upon request. |

For manuscripts utilizing custom algorithms or software that are central to the research but not yet described in published literature, software must be made available to editors and reviewers. We strongly encourage code deposition in a community repository (e.g. GitHub). See the Nature Portfolio guidelines for submitting code & software for further information.

## Data

Policy information about availability of data

All manuscripts must include a data availability statement. This statement should provide the following information, where applicable:
- Accession codes, unique identifiers, or web links for publicly available datasets
- A description of any restrictions on data availability
- For clinical datasets or third party data, please ensure that the statement adheres to our policy

All sequence data has been deposited at NCBI (BioProject PRJNA846736) and the Open Science Framework (https://osf.io/r2un6)

# Field-specific reporting

Please select the one below that is the best fit for your research. If you are not sure, read the appropriate sections before making your selection.

☐ Life sciences  ☐ Behavioural & social sciences  ☒ Ecological, evolutionary & environmental sciences

For a reference copy of the document with all sections, see nature.com/documents/nr-reporting-summary-flat.pdf

# Ecological, evolutionary & environmental sciences study design

All studies must disclose on these points even when the disclosure is negative.

| | |
|---|---|
| Study description | This manuscript reports results from multiple experiments with diverse study designs. Detailed information has been provided in Supplemental Materials. |
| Research sample | This manuscript reports results from a broad spectrum of marine microorganisms. Detailed information has been provided in Supplemental Materials. |
| Sampling strategy | This manuscript reports results from multiple experiments with diverse study designs. Detailed information has been provided in Supplemental Materials. |
| Data collection | This manuscript reports results from multiple experiments with diverse study designs. Detailed information has been provided in Supplemental Materials. |
| Timing and spatial scale | This manuscript reports results from multiple experiments with diverse study designs. Detailed information has been provided in Supplemental Materials. |
| Data exclusions | This manuscript reports results from multiple experiments with diverse study designs. Detailed information has been provided in Supplemental Materials. |
| Reproducibility | All experiments were repeated when possible. |
| Randomization | This manuscript reports results from multiple experiments with diverse study designs. Detailed information has been provided in Supplemental Materials. |
| Blinding | Blinding was not possible, due to the unique nature of each analyzed sample. |

Did the study involve field work?  ☒ Yes  ☐ No

## Field work, collection and transport

| | |
|---|---|
| Field conditions | Detailed information has been provided in Supplemental Materials. |
| Location | Detailed information has been provided in Supplemental Materials. |
| Access & import/export | Seawater samples from the Gulf of Maine were collected off the dock of the Bigelow Laboratory in East Boothbay, Maine, US, and did not require permit. Seawater samples from AT42-11 were collected with consent of the Government of Canada. |
| Disturbance | This study did not cause any environmental disturbance. |

# Reporting for specific materials, systems and methods

We require information from authors about some types of materials, experimental systems and methods used in many studies. Here, indicate whether each material, system or method listed is relevant to your study. If you are not sure if a list item applies to your research, read the appropriate section before selecting a response.

## Materials & experimental systems

| n/a | Involved in the study |
|---|---|
| ☒ ☐ | Antibodies |
| ☒ ☐ | Eukaryotic cell lines |
| ☒ ☐ | Palaeontology and archaeology |
| ☒ ☐ | Animals and other organisms |
| ☒ ☐ | Human research participants |
| ☒ ☐ | Clinical data |
| ☒ ☐ | Dual use research of concern |

## Methods

| n/a | Involved in the study |
|---|---|
| ☒ ☐ | ChIP-seq |
| ☐ ☒ | Flow cytometry |
| ☒ ☐ | MRI-based neuroimaging |

# Flow Cytometry

## Plots

Confirm that:

☒ The axis labels state the marker and fluorochrome used (e.g. CD4-FITC).

☒ The axis scales are clearly visible. Include numbers along axes only for bottom left plot of group (a 'group' is an analysis of identical markers).

☐ All plots are contour plots with outliers or pseudocolor plots.

☐ A numerical value for number of cells or percentage (with statistics) is provided.

## Methodology

| | |
|---|---|
| Sample preparation | Samples of seawater and bacterial cultures were labeled using either SYTO-9 or RedoxSensor Green (Thermo Fisher Scientific) |
| Instrument | Influx Mariner (BD) and ZE5 (Bio-Rad) |
| Software | Sortware and FlowJo (BD) |
| Cell population abundance | An individual cell was collected in each microplate well. The accuracy of droplet deposition into microplate wells was confirmed several times during each sort day by sorting 3.46 μm diameter SPHERO Rainbow Fluorescent Particles (Spherotech, Inc., Lake Forest, IL) and microscopically examining their presence at the bottom of each well. In these examinations, <2% wells did not contain beads and <0.4% wells contained more than one bead. |
| Gating strategy | Cells were gated based on particle green fluorescence (531/40 BP), forward scatter, and the ratio of green versus red fluorescence (692/40 BP - for improved discrimination of cells from detrital particles). Killed negative controls were used to determine RSG background fluorescence. |

☒ Tick this box to confirm that a figure exemplifying the gating strategy is provided in the Supplementary Information.

