## [Peer Review File · Nature]

Manuscript Title: Decoupling of respiration rates and abundance in marine prokaryoplankton

Reviewer Comments & Author Rebuttals

Reviewer Reports on the Initial Version:

Referees' comments:

Referee #1 (Remarks to the Author):

In this manuscript the authors present data from a novel method in which they measure single cell respiration rates. Their results show that the less abundant lineages do most of the respiration and the most abundant heterotrophic bacteria, such as SAR11, are essentially not respiring oxygen. They combine this data with single cell transcriptomics and show that the most inactive heterotrophic bacteria are transcribing proteorhodopsin. They conclude that SAR11 are relying on sunlight to fulfill much of their energy needs. Taxon specific respiration rates, or at least an understanding of the metabolically active pool of bacteria is important to driving parameterizations of models that aim to predict the balance of carbon fixation and respiration in the ocean. So this study is very relevant and a topic of interest to a wide audience.

I am not too well equipped to comment on the technical aspects of the specific techniques used here. But, the authors present a new approach that uses RedoxSensor Green and they spend a good part of this manuscript presenting results from tests aimed at showing that it can be used to measure single cell respiration rates. One could argue that this would best be presented in a separate method paper and that the submitted manuscript could then focus on using the method knowing its limitations. This is a pretty cool tool and I would be thrilled to use it in my own research. Instead they are presenting a new method and their findings from the first application of the method. Some of my comments relate to the method and whether it is actually appropriate to use on small organisms with very low respiration rates and then I also assume the method works and comment on the interpretation of the data.

Did the authors demonstrate that the method works on tiny cells with low respiration rates? I am not sure I am convinced.

In Figure 1, pelagibacter was not used in the calibration of the method. Were any cells used that had respiration rates that were as low as those found in SAR11 in the field samples? Were any as small as SAR11? I ask this because in Figures 3 and 4 when respiration rates are plotted against transcripts and rhodopsin transcripts per cell, it looks to me like there are a whole bunch of data points gathered at the lowest end of the scale for respiration. This to me looks like they may all be at the limit of detection for the method (Figure 4A). Why else would there be such a sharp cutoff with so many data points right on the same exact value? In other words, there is a whole lot of variability in rhodopsin transcripts at the exact same respiration rate. This looks like the respiration rates are below detection or at least below the limit of quantification. As a result, cells with the minimum value of respiration have a wide variety of transcript abundances for 16S rRNA (figure 3D). The

authors draw a lot of their conclusions from the relationship between rhodopsin and respiration and yet I am not convinced that the authors are measuring the respiration adequately at the low end of the scale. One might argue that if the rates are that low, it indicates that the cells are essentially inactive. I think the root of my issue is that in one sentence the authors state that the calibration is from 1-1,000 amol O₂/hr/cell but then the detection threshold is 4amol O₂/hr/cell. How can both be true?

If the detection limit is 4 amol O₂/hr/cell and SAR11 is ~5 amol O₂/hr/cell for the day night study how can the difference between day and night be discerned? It seems like the values are below the limit of reliable quantification. A detection limit is not the same as a quantification limit that allows you to see meaningful differences between cells.

In Figure 1, the fluorescence intensity is the same for values of respiration ranging from 1-4 amol O₂/hr/cell. Given that this is the range of values for SAR11, it doesn't seem like the method really works at that scale. Perhaps I am missing something? I really am hopeful that this method can work on the small cells with low metabolic activity but from the data I am shown I am not certain I can conclude this. Why was SAR11 not used in the calibration? To draw the conclusions drawn in this paper I would like to see this applied to a SAR11 culture first.

In Figure 1b there is almost no data in the low end of the scale and the data that is there suggests that the Winkler method gives much higher respiration rates than the RSG method at the low end. It is not clear to me that the RSG method works very well below about 0.01umolO₂/L/hr. The 95% confidence interval spans about one order of magnitude at that level. This means that telling the difference between 3 and 6 amol/cell/hr is probably impossible. SAR11 seems to always be right at the detection level which is expected given their small size. If they were twice as big they would have 8x the respiration per cell volume as the authors point out. Again, maybe I am misunderstanding and the authors should have a chance to explain this.

It is possible I am misreading the results of these two figures. Let's assume I am, and that the data are sound. I still have some comments on the interpretation of the data and its significance.

The techniques applied in this study are novel and cutting edge and are the reason they are able to probe respiration rates at the single cell level unlike past studies that measure bulk activities like community respiration. I think it is commonly known that some heterotrophic bacterial cells are much more actively respiring than others in the environment. Terms like the "undead" have been used to describe metabolically inactive cells that are viable if put into the right conditions. Likewise, I have heard some describe SAR11 as tiny sacs of inactive cytoplasm just hanging out. But this study shows that in much more detail and with much more refined tools. This is a step forward in our understanding of the role of particular bacterial types in ocean respiration. Combining single cell omics with single cell respiration rate measurements is a very elegant approach and a huge technical feat.

The results gleaned from this method suggest that in fact SAR11 and SAR86 are metabolically inactive. Yet, how then can we explain their abundance? Or prior reports of these taxa being active, doubling, synthesizing proteins and taking up substrates? They must be at least somewhat active or

they would not be the most abundant heterotrophs on earth.

Gomez-Consarnau et al., 2019 quite thoroughly show that rhodopsins are very abundant in the ocean and harvest as much light as Chl-a in the surface ocean. To my knowledge, rhodopsins are capable of harvesting energy from the sun, but in contrast to Chl-a, energy harvesting by rhodopsins is not always coupled to carbon fixation. In fact, rhodopsins are typically not coupled to carbon fixation as is the case in the coupling of Chl and RUBISCO. There are some examples of bacteria that can use PRs to drive carbon fixation through anapleurotic reactions and the glyoxylate shunt (Palovara et al., 2014). In that case, PR based energy capture could result in carbon fixation and subsequent respiration of that fixed carbon - as happens in photoautotrophs. But this is not thought to be the rule and the extent to which it is or is not the rule is not known. To my knowledge, PR is used to make ATP that can fuel flagellar motility, transport across membranes and other functions that promote survival in a low energy environment. I could imagine cells being able to do some basic things through import of organic molecules and ATP supplied by rhodopsin, but I doubt they can grow this way without the TCA cycle churning to make metabolic intermediates for biosynthesis. The data presented here suggests that in fact cells that are using PR are not respiring much leading me to believe that rhodopsins not only are not driving a lot of carbon fixation, but that any organic molecules that may be transported into the cell using ATP from rhodopsins are also not respired. The data suggest that the PR doesn't really help them respire at all. Respiration serves two main functions in the cell – release of energy through the oxidation of organic matter for ATP synthesis and production of biosynthetic intermediates. This second role for respiration is not explicitly stated anywhere in this manuscript and is excluded from figure 4B. The article reads as though acquisition of energy is the sole function of respiration and therefore PR based energy harvesting is a replacement for respiration. Respiration and PR based light harvesting are not at all equal. So what are the cells doing with all this extra ATP they get from the sun? According to the authors, they are not fixing carbon and they are not transporting a lot of materials into the cell - If they were, we would surely see that coupled to respiration and presumably growth. So, at most, this energy is allowing the cells to exist in a static state. If they are not growing and their population is large then this implies no sink for their cells – meaning, that they have no predators. Given their abundance it seems unlikely that they have no predators. So, to me something does not add up. The cells are growing and dividing. Most microbes that we detect in any appreciable numbers in the ocean are growing at their optimal rate or they quickly are lost through cell death and export and become part of the rare biosphere.

Line 35 I agree that these rhodopsins are harvesting energy. I am just not sure this manuscript gets us closer to understanding what they are using this energy for or how it is benefiting them.

Line 52 the authors state that photosynthesizers are pretty well understood but respiration is a black box. They then switch from “microbial respiration” to prokaryoplankton in the next sentence. This seems like a bit of a leap. There are lots of eukaryotes that respire – all of them in fact. We also probably do not totally understand them. Many of them also contain rhodopsins.

Line 188 I am wondering about a BioRxiv article from 2020. Is this published yet? If not why not. If it is then cite the article not the preprint.

Line 243 The authors state that prior work suggests that proteorhodopsin absorbs as much sunlight as Chl-a in the offshore environments. My impression is that both prokaryotes and eukaryotes contribute to this pool of proteorhodopsin, yet the authors here seem to be attributing it all to prokaryotes?

The authors refer to respiration as an “energy source” in several places in the manuscript. I think this wording is a little odd. I think of the substrate as the energy source, e.g. glucose, and respiration is the process that releases the energy to the cell and transforms it in a usable form. Of course respiration provides energy to the cell but is it an energy source?

I think there needs to be more explicit discussion of the relative role in metabolism of proteorhodopsin based energy harvesting vs respiration. The scheme in figure 4B shows these two processes solely as energy yielding reaction. The reality is that respiration provides many more functions to the cell paramount among them being metabolic intermediates. So, proteorhodopsin based energy harvesting is not exactly a replace for respiration, it is an entirely different mode of living. What are they doing with the energy? In an organism in which rhodopsin is creating ATP that is used to fix carbon the organisms is essentially converting into an autotroph. If no carbon fixation happens, the cell still needs organic substrates to come from somewhere and for those substrates to go through the TCA cycle to be useful – unless their environment contains everything they need in the right proportions to grow. Also, what about the nutrients they need? Nitrate and phosphate. How does this play out in the surface ocean where nutrients are limited?

I think Figure 4 looks good but it does not help me see the difference between day and night respiration rates very well. Isn't this the point of this figure? Also, the y-axis is just O₂ consumption. Is that amol? Fmol?

Referee #2 (Remarks to the Author):

Munson-McGee and collaborators propose a study driven by identifying cell-specific quantitative features of prokaryoplankton. In particular, they apply RedoxSensor Green and suggest its use as a proxy for microbial oxygen respiration rate for different cells and environmental contexts. Such application is motivated by the modeling bottleneck for identifying critical parameter values but often out of reach of state-of-the-art parameterization techniques that focus on bulk experiments. Quantitative estimations of respiration differ by orders of magnitude, and Munson-McGee and collaborators suggest complementary studies by performing SAG transcriptome analysis. Conducting both studies is insightful and allows the hypothesis's design (or redefinition) for explaining differences in respiration magnitude. The authors delineate respiration variability with cell size but use proteorhodopsin-based metabolic pathways. Each study is well-conducted and adequately described.

The authors then aim at integrating both results -- which remain a difficult task because of data heterogeneities (i.e., quantitative one for RSG and semi-quantitative for SAG). Munson-McGee and collaborators suggest using significant scores for each SAG experiment and performing the integrating step via correlation analysis. This last section is maybe a weaker stage in the proposed

methodology. First, the authors did not correctly address the compositional nature of transcriptomic data. Second, one could expect a broader genome-scale analysis of SAG's results. The central integrated output correlates single-cell respiration rate and specific gene transcripts. These results are insightful and original, but one could regret results with other annotated genes associated with general intracellular redox metabolic pathways.

As a general comment, I would suggest testing the effect of a clr normalization on transcript results (read counts) before scoring and correlation analysis. It would unbiased correlation analysis between quantitative and semi-quantitative data. Complementary, I would suggest applying state-of-the-art network analysis to delineate a set of highly co-expressed genes with respiration rates. I would also recommend performing metabolic network reconstruction from SAG transcript -- when annotations are accessible -- to compare quantitative metabolic results with single-cell respiration rates. Such modeling allows the estimation of the overall respiration under quasi-steady-state assumptions. It could be critical to understanding further identified genome-scale uncertainties and their relationship with novel quantitative single-cell rates.

Aylward, F. O., Eppley, J. M., Smith, J. M., Chavez, F. P., Scholin, C. A., & DeLong, E. F. (2015). Microbial community transcriptional networks are conserved in three domains at ocean basin scales. *Proceedings of the National Academy of Sciences of the United States of America*, 112(17), 5443–5448. <https://doi.org/10.1073/pnas.1502883112>

Machado, D., Andrejev, S., Tramontano, M., & Patil, K. R. (2018). Fast automated reconstruction of genome-scale metabolic models for microbial species and communities. *Nucleic Acids Research*, 46(15), 7542–7553. <https://doi.org/10.1093/nar/gky537>

Referee #3 (Remarks to the Author):

I have reviewed manuscript 2022-02-02239 by Stepanauskas et al. The authors report rates of cell-specific respiration of microbial cells from the Gulf of Maine that they found to differ by more than a factor of 1000. They report that a majority of the respiration was associated with cells that made up a minority of the prokaryoplankton population and that the respiration rates of a majority of the cells were extremely low. They found elevated counts of proterorhodopsin transcripts in some of the cells whose respiration rates were extremely low. They conclude that proterorhodopsin-based heterotrophy is probably an important source of energy for prokaryoplankton.

The principal problem I see with this analysis is that the authors implicitly assume that all of the cells they studied were actively growing or at least growing at rates that would be associated with requirements for energy much greater than the rates implied by the results of their respiration measurements. Large numbers of cells do not necessarily imply rapid growth. I recall a conversation I had with Richard Barber years ago concerning phytoplankton in the Peruvian upwelling system. According to Dr. Barber, the cells in recently upwelled water were very weakly pigmented and appeared anemic under the microscope but were growing fixing carbon very rapidly. At the end of the bloom, the cells were full of pigments and appeared healthy but were fixing carbon at very low rates because essential nutrients had been stripped from the water. Without a quantitative

assessment of growth rates, it is impossible to say whether the abundant cells with low respiration rates were growing at rates that would logically be associated with large requirements for energy or not. An alternative scenario is that the abundant cells were the remnants of a bloom and were in fact growing very slowly if at all. Without independent measurements of growth rates, I think it is premature to hypothesize that the abundant cells were obtaining much of their energy via proterorhodopsin-based heterotrophy.

Author Rebuttals to Initial Comments:

Response to reviewers, Nature manuscript 2022-02-02239:

Decoupling of respiration rates and abundance in marine prokaryoplankton

Jacob H. Munson-McGee, Melody R. Lindsay, Julia M. Brown, Eva Sintes, Timothy D'Angelo, Joe Brown, Laura C. Lubelczyk, Paxton Tomko, David Emerson, Beth N. Orcutt, Nicole J. Poulton, Gerhard J. Herndl, and Ramunas Stepanauskas

Referees' comments:

Referee #1 (Remarks to the Author):

In this manuscript the authors present data from a novel method in which they measure single cell respiration rates. Their results show that the less abundant lineages do most of the respiration and the most abundant heterotrophic bacteria, such as SAR11, are essentially not respiring oxygen. They combine this data with single cell transcriptomics and show that the most inactive heterotrophic bacteria are transcribing proteorhodopsin. They conclude that SAR11 are relying on sunlight to fulfill much of their energy needs. Taxon specific respiration rates, or at least an understanding of the metabolically active pool of bacteria is important to driving parameterizations of models that aim to predict the balance of carbon fixation and respiration in the ocean. So this study is very relevant and a topic of interest to a wide audience.

I am not too well equipped to comment on the technical aspects of the specific techniques used here. But, the authors present a new approach that uses RedoxSensor Green and they spend a good part of this manuscript presenting results from tests aimed at showing that it can be used to measure single cell respiration rates. One could argue that this would best be presented in a separate method paper and that the submitted manuscript could then focus on using the method knowing its limitations. This is a pretty cool tool and I would be thrilled to use it in my own research. Instead they are presenting a new method and their findings from the first application of the method. Some of my comments relate to the method and whether it is actually appropriate to use on small organisms with very low respiration rates and then I also assume the method works and comment on the interpretation of the data.

Did the authors demonstrate that the method works on tiny cells with low respiration rates? I am not sure I am convinced.

In Figure 1, pelagibacter was not used in the calibration of the method. Were any cells used that had respiration rates that were as low as those found in SAR11 in the field samples? Where any as small as SAR11? I ask this because in Figures 3 and 4 when respiration rates are plotted against transcripts and rhodopsin transcripts per cell, it looks to me like there are a whole bunch of data points gathered at the lowest end of the scale for respiration. This to me looks like they may all be at the limit of detection for the method (Figure 4A). Why else

would there be such a sharp cutoff with so many data points right on the same exact value? In other words, there is a whole lot of variability in rhodopsin transcripts at the exact same respiration rate. This looks like the respiration rates are below detection or at least below the limit of quantification. As a result, cells with the minimum value of respiration have a wide variety of transcript abundances for 16S rRNA (figure 3D).

We thank the reviewer for their thoughtful feedback. In response to the comment above regarding respiration calibration (Figure 1A), we want to point out that multiple cultures in our respiration calibration had respiration rates between 2-7 $\text{amol O}_2 \text{ cell}^{-1} \text{ hr}^{-1}$, i.e. range matching the $\sim 5 \text{ amol O}_2 \text{ cell}^{-1} \text{ hr}^{-1}$ respiration estimated for *Pelagibacter* in this manuscript and in an earlier, cultivation-based study (Steindler et al. 2011). Furthermore, one of our calibration cultures, *Rhodoluna lacicola*, had $\sim 0.4 \mu\text{m}$ cell diameter (Supplemental Table 1), similar to the $\sim 0.3 \mu\text{m}$ diameter of *Pelagibacter* cells. Thus, we believe that our calibration efforts adequately represent the range of respiration rates and cell sizes of the majority of prokaryoplankton lineages in our field samples.

In response to the comment regarding respiration measurements in field samples (Figures 2-4), we would like to provide further clarification. Please note that the genus-specific respiration estimates take into account cells that fell below the RSG detection limit, as described in Materials and Methods, lines 813-834. Specifically, cells below detection limit were assigned respiration rates equal to $\frac{1}{2}$ of the detection limit, which is a standard practice. As a result, all genera with no or few cells above detection limit were assigned respiration rate values equal or slightly above $\frac{1}{2}$ of the detection limit. This has been noted in the captions of Figures 2- 4.

The authors draw a lot of their conclusions from the relationship between rhodopsin and respiration and yet I am not convinced that the authors are measuring the respiration adequately at the low end of the scale. One might argue that if the rates are that low, it indicates that the cells are essentially inactive. I think the root of my issue is that in one sentence the authors state that the calibration is from 1-1,000 $\text{amol O}_2/\text{hr}/\text{cell}$ but then the detection threshold is $4 \text{ amol O}_2/\text{hr}/\text{cell}$. How can both be true?

We determined our detection threshold in seawater as $4 \text{ amol O}_2 \text{ cell}^{-1} \text{ hr}^{-1}$ based on extensive gating efforts on live field samples and background fluorescence in killed controls, as described on lines 88-89. This threshold was used in all analyses of field samples. Background fluorescence was lower in laboratory cultures, allowing us to compare RSG and optode measurements in a slightly broader range, 1-1,000 $\text{amol O}_2 \text{ cell}^{-1} \text{ hr}^{-1}$, as reported in Fig. 1A. Please note that we did not use $1 \text{ amol O}_2 \text{ cell}^{-1} \text{ hr}^{-1}$ as a detection limit in any of our analyses.

If the detection limit is $4 \text{ amol O}_2/\text{hr}/\text{cell}$ and SAR11 is $\sim 5 \text{ amol O}_2/\text{hr}/\text{cell}$ for the day night study how can the difference between day and night be discerned? It seems like the values are below the limit of reliable quantification. A detection limit is not the same as a quantification limit that allows you to see meaningful differences between cells.

We agree and do not describe any differences in respiration rate between SAR11 cells between the day and night samples. This is explicitly stated on lines 232-233. We do report a day-night respiration difference for SAR86 cells, which respired $\sim 8 \text{ amol O}_2 \text{ cell}^{-1} \text{ hr}^{-1}$ during the day and $52 \text{ amol O}_2 \text{ cell}^{-1} \text{ hr}^{-1}$ during the night. This 6x difference is substantially above the detection limit. In the revised manuscript, we corrected and further clarified the difference in SAR86 respiration rates between day and night measurements, lines 231-234.

In Figure 1, the fluorescence intensity is the same for values of respiration ranging from 1-4 $\text{amol O}_2/\text{hr}/\text{cell}$. Given that this is the range of values for SAR11, it doesn't seem like the method really works at that scale. Perhaps I am missing something? I really am hopeful that this method can work on the small cells with low metabolic activity but from the data I am shown I am not certain I can conclude this. Why was SAR11 not used in the calibration? To draw the conclusions drawn in this paper I would like to see this applied to a SAR11 culture first.

While we did not use SAR11 in the calibration curve, due to culture availability challenges, we did compare our environmentally measured respiration rates to respiration rates that have previously been measured in cultures of SAR11. Our average measured SAR11 respiration rate of $\sim 5 \text{ amol O}_2 \text{ cell}^{-1} \text{ hr}^{-1}$ is within the 1-10 $\text{amol O}_2 \text{ cell}^{-1} \text{ hr}^{-1}$ range that has previously been measured for SAR11 cells in culture (Steindler et. Al. 2011), as reported on manuscript line 143.

In Figure 1b there is almost no data in the low end of the scale and the data that is there suggests that the Winkler method gives much higher respiration rates than the RSG method at the low end. It is not clear to me that the RSG method works very well below about $0.01 \mu\text{mol O}_2/\text{L}/\text{hr}$. The 95% confidence interval spans about one order of magnitude at that level. This means that telling the difference between 3 and 6 $\text{amol}/\text{cell}/\text{hr}$ is probably impossible. SAR11 seems to always be right at the detection level which is expected given their small size. If they were twice as big they would have 8x the respiration per cell volume as the authors point out. Again, maybe I am misunderstanding and the authors should have a chance to explain this.

We agree with the reviewer that the observed SAR11 per-cell respiration rate is at the verge of our detection limit and point it out in multiple places in the manuscript, e.g. lines 136-138 and 164-165.

It is possible I am misreading the results of these two figures. Let's assume I am, and that the data are sound. I still have some comments on the interpretation of the data and its significance.

The techniques applied in this study are novel and cutting edge and are the reason they are able to probe respiration rates at the single cell level unlike past studies that measure bulk activities like community respiration. I think it is commonly known that some heterotrophic bacterial cells are much more actively respiring than others in the environment. Terms like the "undead" have been used to describe metabolically inactive cells that are viable if put into the right conditions. Likewise, I have heard some describe SAR11 as tiny sacs of inactive cytoplasm just hanging out. But this study shows that in much more detail and with

much more refined tools. This is a step forward in our understanding of the role of particular bacterial types in ocean respiration. Combining single cell omics with single cell respiration rate measurements is a very elegant approach and a huge technical feat.

We appreciate this positive assessment of our manuscript.

The results gleaned from this method suggest that in fact SAR11 and SAR86 are metabolically inactive. Yet, how then can we explain their abundance? Or prior reports of these taxa being active, doubling, synthesizing proteins and taking up substrates? They must be at least somewhat active or they would not be the most abundant heterotrophs on earth.

Our results suggest that SAR11, SAR86 and many other marine prokaryoplankton lineages have low (but not zero) per-cell respiration rates. We do not suggest that these lineages are metabolically inactive, because respiration is only one of many forms of microbial cellular metabolism. In fact, our metatranscriptomic data indicate metabolic activity in all analyzed lineages, and we specifically suggest that rhodopsin phototrophy - another type of energy metabolism - may to some extent compensate for the low rate of respiration. Furthermore, while SAR86 cells respired at low rates during the day, we found a 6x increase in their respiration rate during the night.

Gomez-Consarnau et al., 2019 quite thoroughly show that rhodopsins are very abundant in the ocean and harvest as much light as Chl-a in the surface ocean. To my knowledge, rhodopsins are capable of harvesting energy from the sun, but in contrast to Chl-a, energy harvesting by rhodopsins is not always coupled to carbon fixation. In fact, rhodopsins are typically not coupled to carbon fixation as is the case in the coupling of Chl and RUBISCO. There are some examples of bacteria that can use PRs to drive carbon fixation through anapleurotic reactions and the glyoxylate shunt (Palovara et al., 2014). In that case, PR based energy capture could result in carbon fixation and subsequent respiration of that fixed carbon - as happens in photoautotrophs. But this is not thought to be the rule and the extent to which it is or is not the rule is not known. To my knowledge, PR is used to make ATP that can fuel flagellar motility, transport across membranes and other functions that promote survival in a low energy environment. I could imagine cells being able to do some basic things through import of organic molecules and ATP supplied by rhodopsin, but I doubt they can grow this way without the TCA cycle churning to make metabolic intermediates for biosynthesis. The data presented here suggests that in fact cells that are using PR are not respiring much leading me to believe that rhodopsins not only are not driving a lot of carbon fixation, but that any organic molecules that may be transported into the cell using ATP from rhodopsins are also not respired. The data suggest that the PR doesn't really help them respire at all. Respiration serves two main functions in the cell – release of energy through the oxidation of organic matter for ATP synthesis and production of biosynthetic intermediates. This second role for respiration is not explicitly stated anywhere in this manuscript and is excluded from figure 4B. The article reads as though acquisition of energy is the sole function of respiration and therefore PR based energy harvesting is a replacement for respiration. Respiration and PR based light harvesting are not at all equal.

So what are the cells doing with all this extra ATP they get from the sun? According to the authors, they are not fixing carbon and they are not transporting a lot of materials into the cell - If they were, we would surely see that coupled to respiration and presumably growth. So, at most, this energy is allowing the cells to exist in a static state. If they are not growing and their population is large then this implies no sink for their cells – meaning, that they have no predators. Given their abundance it seems unlikely that they have no predators. So, to me something does not add up. The cells are growing and dividing. Most microbes that we detect in any appreciable numbers in the ocean are growing at their optimal rate or they quickly are lost through cell death and export and become part of the rare biosphere.

The reviewer raises a good point about the “second” function of respiration regarding the production of biosynthetic intermediaries. However, we feel that analyses of biosynthetic metabolisms are beyond the scope of this manuscript. Figure 4B explicitly states that it is focused on the generation of a proton gradient and ATP in prokaryoplankton and not on other metabolisms. To clarify further, we added the following statement on lines 210-211: “Thus, some prokaryoplankton lineages may be mixotrophs that rely on a combination of heterotrophic and phototrophic processes as energy sources.”.

Line 35 I agree that these rhodopsins are harvesting energy. I am just not sure this manuscript gets us closer to understanding what they are using this energy for or how it is benefiting them.

We agree with the reviewer that there is a lot remaining to be understood about the role of rhodopsins in marine prokaryoplankton.

Line 52 the authors state that photosynthesizers are pretty well understood but respiration is a black box. They then switch from “microbial respiration” to prokaryoplankton in the next sentence. This seems like a bit of a leap. There are lots of eukaryotes that respire – all of them in fact. We also probably do not totally understand them. Many of them also contain rhodopsins.

We have clarified this sentence to specify that we are focusing on bacterial and archaeal respiration.

Line 188 I am wondering about a BioRxiv article from 2020. Is this published yet? If not why not. If it is then cite the article not the preprint.

We thank the reviewer for noticing this and have updated the reference accordingly.

Line 243 The authors state that prior work suggests that proteorhodopsin absorbs as much sunlight as Chl-a in the offshore environments. My impression is that both prokaryotes and

eukaryotes contribute to this pool of proteorhodopsin, yet the authors here seem to be attributing it all to prokaryotes?

We agree with the reviewer that the role of rhodopsins in eukaryotic plankton remains poorly understood, although it is generally assumed that bacteria and archaea dominate the pool of rhodopsins in the epipelagic. On line 243, our reference to prior literature does not imply any assumptions about the relative distribution of rhodopsins among lineages of life.

The authors refer to respiration as an “energy source” in several places in the manuscript. I think this wording is a little odd. I think of the substrate as the energy source, e.g. glucose, and respiration is the process that releases the energy to the cell and transforms it in a usable form. Of course respiration provides energy to the cell but is it an energy source?

We agree with the reviewer’s comment and changed “energy source” to “energy producing process” in reference to respiration.

I think there needs to be more explicit discussion of the relative role in metabolism of proteorhodopsin based energy harvesting vs respiration. The scheme in figure 4B shows these two processes solely as energy yielding reaction. The reality is that respiration provides many more functions to the cell paramount among them being metabolic intermediates. So, proteorhodopsin based energy harvesting is not exactly a replace for respiration, it is an entirely different mode of living. What are they doing with the energy? In an organism in which rhodopsin is creating ATP that is used to fix carbon the organisms is essentially converting into an autotroph. If no carbon fixation happens, the cell still needs organic substrates to come from somewhere and for those substrates to go through the TCA cycle to be useful – unless their environment contains everything they need in the right proportions to grow. Also, what about the nutrients they need? Nitrate and phosphate. How does this play out in the surface ocean where nutrients are limited?

As stated above, the reviewer raises a good point about the “second” function of respiration regarding the production of biosynthetic intermediaries. However, we feel that analyses of biosynthetic metabolisms are beyond the scope of this manuscript. Figure 4B explicitly states that it is focused on the generation of a proton gradient and ATP in prokaryoplankton and not on other metabolisms. To clarify further, we added the following statement on lines 210-211: “Thus, some prokaryoplankton lineages may be mixotrophs that rely on a combination of heterotrophic and phototrophic processes as energy sources.”.

I think Figure 4 looks good but it does not help me see the difference between day and night respiration rates very well. Isn’t this the point of this figure? Also, the y-axis is just O₂ consumption. Is that amol? Fmol?

We thank the reviewer for this comment and have revised the y-axis accordingly.

Referee #2 (Remarks to the Author):

Munson-McGee and collaborators propose a study driven by identifying cell-specific quantitative features of prokaryoplankton. In particular, they apply RedoxSensor Green and suggest its use as a proxy for microbial oxygen respiration rate for different cells and environmental contexts. Such application is motivated by the modeling bottleneck for identifying critical parameter values but often out of reach of state-of-the-art parameterization techniques that focus on bulk experiments.

Quantitative estimations of respiration differ by orders of magnitude, and Munson-McGee and collaborators suggest complementary studies by performing SAG transcriptome analysis. Conducting both studies is insightful and allows the hypothesis's design (or redefinition) for explaining differences in respiration magnitude. The authors delineate respiration variability with cell size but use proteorhodopsin-based metabolic pathways. Each study is well-conducted and adequately described.

The authors then aim at integrating both results -- which remain a difficult task because of data heterogeneities (i.e., quantitative one for RSG and semi-quantitative for SAG). Munson-McGee and collaborators suggest using significant scores for each SAG experiment and performing the integrating step via correlation analysis. This last section is maybe a weaker stage in the proposed methodology. First, the authors did not correctly address the compositional nature of transcriptomic data. Second, one could expect a broader genome-scale analysis of SAG's results. The central integrated output correlates single-cell respiration rate and specific gene transcripts. These results are insightful and original, but one could regret results with other annotated genes associated with general intracellular redox metabolic pathways.

As a general comment, I would suggest testing the effect of a clr normalization on transcript results (read counts) before scoring and correlation analysis. It would unbiased correlation analysis between quantitative and semi-quantitative data. Complementary, I would suggest applying state-of-the-art network analysis to delineate a set of highly co-expressed genes with respiration rates. I would also recommend performing metabolic network reconstruction from SAG transcript -- when annotations are accessible -- to compare quantitative metabolic results with single-cell respiration rates. Such modeling allows the estimation of the overall respiration under quasi-steady-state assumptions. It could be critical to understanding further identified genome-scale uncertainties and their relationship with novel quantitative single-cell rates.

We thank the reviewer for their feedback and thoughts on the study. Regarding their suggestion of testing a clr normalization, however, we feel that it is unnecessary. The reviewer states that this normalization would unbiased the correlation between quantitative (read counts) and semi-quantitative data (SAGs). However, we did not use the abundance of SAGs when calculating the number of cells $\text{genus}^{-1} \text{ ml}^{-1}$. Instead, we used the recruitment of metagenomic reads on SAGs of each genus as a reference database, which is a robust,

quantitative technique and as such makes this normalization unnecessary, see lines 788-810.

While metabolic modeling would obviously be informative, it is well beyond the scope of this manuscript. Accurate metabolic modeling of a single species can be an extremely challenging and time-consuming process, while this study is focused on processes at the level of entire prokaryoplankton community.

Aylward, F. O., Eppley, J. M., Smith, J. M., Chavez, F. P., Scholin, C. A., & DeLong, E. F. (2015). Microbial community transcriptional networks are conserved in three domains at ocean basin scales. *Proceedings of the National Academy of Sciences of the United States of America*, 112(17), 5443–5448. <https://doi.org/10.1073/pnas.1502883112>

Machado, D., Andrejev, S., Tramontano, M., & Patil, K. R. (2018). Fast automated reconstruction of genome-scale metabolic models for microbial species and communities. *Nucleic Acids Research*, 46(15), 7542–7553. <https://doi.org/10.1093/nar/gky537>

Referee #3 (Remarks to the Author):

I have reviewed manuscript 2022-02-02239 by Stepanauskas et al. The authors report rates of cell-specific respiration of microbial cells from the Gulf of Maine that they found to differ by more than a factor of 1000. They report that a majority of the respiration was associated with cells that made up a minority of the prokaryoplankton population and that the respiration rates of a majority of the cells were extremely low. They found elevated counts of proterorhodopsin transcripts in some of the cells whose respiration rates were extremely low. They conclude that proterorhodopsin-based heterotrophy is probably an important source of energy for prokaryoplankton.

The principal problem I see with this analysis is that the authors implicitly assume that all of the cells they studied were actively growing or at least growing at rates that would be associated with requirements for energy much greater than the rates implied by the results of their respiration measurements. Large numbers of cells do not necessarily imply rapid growth. I recall a conversation I had with Richard Barber years ago concerning phytoplankton in the Peruvian upwelling system. According to Dr. Barber, the cells in recently upwelled water were very weakly pigmented and appeared anemic under the microscope but were growing fixing carbon very rapidly. At the end of the bloom, the cells were full of pigments and appeared healthy but were fixing carbon at very low rates because essential nutrients had been stripped from the water. Without a quantitative assessment of growth rates, it is impossible to say whether the abundant cells with low respiration rates were growing at rates that would logically be associated with large requirements for energy or not. An alternative scenario is that the abundant cells were the remnants of a bloom and were in fact growing very slowly if at all. Without independent measurements of growth rates, I think it is premature to hypothesize that the abundant cells were obtaining much of their energy via proterorhodopsin-based heterotrophy.

We thank the reviewer for their comments. We would like to emphasize that we did not measure prokaryoplankton growth rates and made no assumptions about growth rates in the analyzed samples. This manuscript reports, for the first time, the rates of respiration at the resolution of individual prokaryoplankton cells.

The reviewer is right that some microbial lineages may have a boom-and-bust demographic patterns. However, our data provide strong evidence for respiration rates being consistent over broad temporal (Gulf of Maine studies spanning nearly three years) and spatial (coastal versus offshore) scales among lineages that are well represented in our dataset (e.g. *Pelagibacter*, SAR86, *Roseobacter*). We modified sentences on lines 138, and 178-179 to make this clearer. Our findings are consistent with prior literature cited in the manuscript. For example, the multitude of prior reports on *Pelagibacter*, based on both field analyses and laboratory experimentation, have never found blooms. Instead, they consistently suggest an extremely steady rate of metabolism and very limited regulation of gene expression, as cited on lines 175-180.

Reviewer Reports on the First Revision:

Referees' comments:

Referee #1 (Remarks to the Author):

This revised version is an improvement over the prior version. I have only a few comments and if they can be addressed then I would support publication.

Line 199 I am not sure I understand this. Among what genera? Why are there 2 ranges?

Line 206 I may be wrong about this but isn't there a need for resources other than energy to drive transcription? Keeping mRNA high in a cell must also require C, N, and P? Vitamins. Can a cell keep total mRNA high with only energy input? Also, why make the mRNA if it won't be translated into protein. To make protein the cell will need amino acids. Are these coming from the surrounding DOM?

Line 250 I think I would like a slightly more elaborated discussion of what a cell can do with only ATP at its disposal. Is the cell also using external resources? Are these all readily available? Like free amino acids or nitrogenous bases, etc.

Referee #2 (Remarks to the Author):

I thank the authors for their clarification and the second version of the manuscript. I understand the arguments. However, I would still recommend presenting the distribution of the SAG read count to certify the interest of the correlation score better to compare heterogeneous data.

I also understand that metabolic modeling is beyond the scope of the study. But as raised by another reviewer, I would nevertheless recommend better presenting the SAG result (metabolic reconstruction could be a way to emphasize essential genes for respiration - without entirely performing quantitative modeling which is indeed not the point herein). To clarify my concern, I would recommend showing in already abundant supplementary materials the most critical transcripts per genera (and maybe showing the rhodopsin as part of the core via upset diagram or other accurate representations). I am very interested in the study -- but the current form does give credit to the SAG approach and its comparison with the respiration experiments. Such a demonstration could avoid the "cherry-picking" feeling of focusing on the few transcripts of interest.

Referee #3 (Remarks to the Author):

I have reviewed the revised version of manuscript 2022-0202239A by Munson-McGee et al. I think the argument that *Pelagibacter*, SAR86, and perhaps other members of the prokaryoplankton use proteorhodopsin to produce ATP could be strengthened if the authors cited some papers that are

currently missing from the reference list:

1. B  j  , O., L. Aravind, E. V. Koonin, M. T. Suzuki, A. Hadd, L. P. Nguyen, S. B. Jovanovich, C. Gates, R. A. Feldman, J. L. Spudich, E. N. Spudich, and E. F. DeLong. 2000. Bacterial rhodopsin: evidence for a new type of phototrophy in the sea. *Science* 289: 1902-1906.
2. Sabehi, G., B  j  , O., Suzuki, M.T., Preston, C.M. & E. F. DeLong. 2004. Different SAR86 subgroups harbour divergent proteorhodopsins, *Environ. Microbiol.* 6:903–910.
3. A. Martinez, A. Bradley, J. Waldbauer, R. Summons and E. F. DeLong. 2007. Proteorhodopsin photosystem gene expression enables photophosphorylation in a heterologous host. *Proc. Natl Acad Sci.* 104: 5590-5595.
4. DeLong, E. F. and B  j  , O. (2010). The light-driven proton pump proteorhodopsin enhances bacterial survival during tough times. *PLoS Biol* 9: e1000359.
5. Dupont, C. L. et al. 2012. Genomic insights to SAR86, an abundant and uncultivated marine bacterial lineage. *ISME Journal* 6: 1186–1199.

I think these papers collectively make a very strong case that proteorhodopsin is involved in the production of ATP by some prokaryoplankton.

This was, to me, the only weak aspect of the original manuscript. The use of single-cell RSG fluorescence to estimate the respiration rates of single cells is certainly an exciting and potentially very important development. Scientific research is invariably methods limited to some extent, and this is a tool that will enable research that would have been impossible with any other methodology. I see no issues with the data & methodology or the use of statistics. I think the conclusions would be strengthened by citing the above references and information in those references.

Referee #4 (Remarks to the Author):

The manuscript “Decoupling of respiration rates and abundance in marine prokaryoplankton” highlights results from an exciting new method designed to tackle a long standing problem, which is to quantify respiration rates in diverse lineages of uncultured marine bacteria. The justification for this work is well established and the authors do a nice job highlighting the significance of opening the black box of bulk respiration. The authors present a proteorhodopsin-based explanation to support their key observation, which is that the most abundant organisms in the ocean contribute very little to respiration and that rare organisms contribute to the bulk of respiration. Overall this is a very thorough and well written manuscript that introduces an exciting new method and that presents some new results. I have some concerns regarding the focus and some question regarding the method.

Concern regarding the focus:

My primary concern is that the title and key observation highlighting the decoupling of key parameters will be interpreted by some to mean that the most abundant organisms in the ocean are inactive (as noted by previous reviewers). An alternative explanation is that the most abundant lineages of bacteria in the ocean are also the most efficient and adjust carbon and energy sources to maintain high bacterial growth efficiency ($BGE = (BP)/(BP + BR)$), which is calculated with bacterial production not abundance. In their response to reviewers, the authors correctly note that the cells are not inactive, but that light-driven ATP production could provide a significant source of energy. There are other explanations that may contribute and I think that the authors should consider them as well. For example, the glyoxylate shunt, used by SAR11 and other aerobic heterotrophs, bypasses key CO₂ producing steps in the TCA cycle to metabolize specific compounds like fatty acids and two carbon compounds, and to regenerate key intermediates for biosynthesis. Also, in a relatively recent review of SAR11, Giovannoni notes that SAR11 do not use a wide range of energy rich carbohydrates. Having mechanisms that conserve carbon and energy, like the glyoxylate shunt and proteorhodopsin, makes sense in a carbon and energy limited systems, especially if the goal is to maintain high BGE. In short, suggesting that the dominant lineages don't contribute much to respiration will likely propagate the misleading notion that evolutionary success is achieved through dormancy. I encourage the authors to consider focusing on high and low BGE as an alternative.

Reservations regarding the focus:

Second but related, I don't think that decoupled abundance and respiration is surprising. The authors note that low respiration rates have been reported in cultured SAR11 (Steindler et al., 2011). Previous studies have also found that SAR11 don't use a wide range of energy rich carbohydrates but instead use compounds like VOCs, organic acids, phosphonates, polyamines, and amino acids (Giovannoni 2017). It has been known for decades that members of the Roseobacter are often the most abundant cells in phytoplankton blooms and members of this lineage are capable of using a broad range of energy rich carbohydrates. The authors point this out in the manuscript, noting that their data are in agreement with previous work. The ability to measure lineage-specific differences in respiration will fill a critical gap in knowledge and has great potential to transform our understanding of how different lineages have adapted to life in the oceans. But given what is already known about the metabolism of oligotrophs and copiotrophs, I think the novelty is in the development of the method and its application to field samples, not the decoupling of abundance and respiration.

Other significant findings:

Light driven ATP production in SAR11 and other oligotrophic marine bacteria is a potential source of energy, and to me a more interesting result. As the authors note, growth experiments have not shown that marine bacteria with proteorhodopsin have a significant advantage in the light versus the dark. As the authors have shown, respiration in light and dark samples might provide a better answer to how cells benefit, beyond what is known about the benefits associated with declining respiration when cells enter a non-growth state (Steindler et al., 2011). When more energy from the sun is available cells could respire less to maintain high BGE and to conserve carbon, but when less energy is available (no sun) cells could respire more, as the authors suggest. This section is a nice addition to the paper and represents a more novel discovery. Of course, doing this under controlled conditions with an isolate would be ideal and although not feasible for this study, SAR11 isolates are

available from several laboratories upon request.

Questions about the method:

The organisms used to calibrate and validate the RSG-based prokaryoplankton respiration measurements all grow to very high cell densities and on nutrient rich media. They also consume large amounts of oxygen. I agree with other reviewers that this manuscript would be stronger if RSG-based respiration was measured in more representative bacteria and ideally under different growth conditions. In the primary RSG paper referenced by the authors (Kalyuznaya et al., 2008), they used starved cells to show a loss of RSG signal and then added substrate to show increases in RSG signal. In the present study, a loss of RSG signal is only shown for killed controls, which were autoclaved for 30 minutes. This seems a bit harsh. Experiments with representative cultures and with controls under different conditions or that have been starved and/or inactivated, rather than severely altered by autoclaving, would be more compelling. Ideally, it would be nice to have a range of relativized fluorescence intensities for representative organisms. As is, the authors are comparing respiration of low abundance slow growing cells in low nutrient seawater to high abundance fast growing cells in high nutrient media. More specific questions about this are below.

In Figure 1 panel A, the authors plot normalized RSG fluorescence intensity against the rate of oxygen consumption, reported as respiration rate (fmol/cell/hour). In panel B, the authors plot RSG- and Winkler-based respiration measurements from natural samples, but with RSG-based respiration (umol oxygen/L/hour). I assume the difference in units is for direct comparison with the Winkler-based method but would like to know why. Why is panel A in fmol/cell/hour and panel B in umol oxygen/L/hour? Isn't relative fluorescence intensity used to calculate RSG-based respiration in panel B, ie. by using the correlation in panel A? If the points in panel B were added to panel A, using average relative fluorescence intensity values, where would they plot on the y axis (fmol/cell/hour)? If relative fluorescence intensity is low for the natural samples (0-20) then the per cell values are in the range of data that are only represented on part of the graph and that don't look correlated if the higher values obtained from non-marine organisms are removed. I looked for relative fluorescence intensity values in the tables but couldn't find them. Is oxygen consumption comparable between the cultures and the field samples?

In Figure 2 panel D the y axis is "per cell respiration rate (fmol oxygen/hour)". Is this the same as the y axis in Figure 1 panel A "respiration rate (fmol/cell/hour)"? And in Figure 4 panels C and D the y axis is "oxygen consumption amol/cell/hour." Is this the same as respiration rate in Figure 1 panel A, per cell respiration rate in Figure 2 panel D, or are they all the same? Consistency would help with interpretation. Based on these other plots and discussions in the text, most natural samples have low relative fluorescence values.

Author Rebuttals to First Revision:

Dear Dr Caputa,

We appreciate the additional, generally positive feedback from the reviewers. Please see below our detailed responses to the reviewers' comments.

Referees' comments:

Referee #1 (Remarks to the Author):

This revised version is an improvement over the prior version. I have only a few comments and if they can be addressed then I would support publication.

Line 199 I am not sure I understand this. Among what genera? Why are there 2 ranges? The two ranges are for the number of mRNA and rRNA transcripts. To clarify, we rephrased and split this statement into two sentences: "The overall average counts of mRNA and 16S rRNA gene transcripts per cell were 36 and 426, which is similar to the earlier reports on coastal prokaryoplankton³⁰. However, our study showed that these counts varied widely among genera, ranging 5-144 cell⁻¹ for mRNA and 48-2,480 cell⁻¹ for rRNA".

Line 206 I may be wrong about this but isn't there a need for resources other than energy to drive transcription? Keeping mRNA high in a cell must also require C, N, and P? Vitamins. Can a cell keep total mRNA high with only energy input? Also, why make the mRNA if it won't be translated into protein. To make protein the cell will need amino acids. Are these coming from the surrounding DOM?

Line 250 I think I would like a slightly more elaborated discussion of what a cell can do with only ATP at its disposal. Is the cell also using external resources? Are these all readily available? Like free amino acids or nitrogenous bases, etc.

These two comments are related. We agree with the reviewer that energy in the form ATP is only one of a multitude of cellular needs experienced by marine prokaryoplankton. While direct analyses of such needs are outside the scope of our manuscript, we have addressed the reviewer's comment by adding the following statement on lines 216-219: "While it is well-known that *Pelagibacter* and other predominant groups of surface ocean prokaryoplankton consume simple organic compounds, access to proteorhodopsin-derived ATP may increase the use of these molecules in biosynthesis as opposed to respiration, thus increasing growth efficiency". We agree that this addition will improve the clarity of our manuscript to a broader range of *Nature* readers.

Referee #2 (Remarks to the Author):

I thank the authors for their clarification and the second version of the manuscript. I understand the arguments. However, I would still recommend presenting the distribution of the SAG read count to certify the interest of the correlation score better to compare heterogeneous data.

Genomic reads from single amplified genomes (SAGs) were used only to obtain SAG genome assemblies. Read counts are presented in supplemental table 3. It is unclear to us what correlations the reviewer is suggesting to add to the manuscript, and for what purpose.

I also understand that metabolic modeling is beyond the scope of the study. But as raised by another reviewer, I would nevertheless recommend better presenting the SAG result (metabolic reconstruction could be a way to emphasize essential genes for respiration - without entirely performing quantitative modeling which is indeed not the point herein). To clarify my concern, I would recommend showing in already abundant supplementary materials the most critical transcripts per genera (and maybe showing the rhodopsin as part of the core via upset diagram or other accurate representations). I am very interested in the study -- but the current form does give credit to the SAG approach and its comparison with the respiration experiments. Such a demonstration could avoid the "cherry-picking" feeling of focusing on the few transcripts of interest.

We appreciate the comment. We chose to analyze and display rhodopsin transcript counts because of the specific focus of our study. Analyzing the full metabolic complexity marine prokaryoplankton would be well outside the scope of this manuscript, while choosing "the most critical transcripts per genera" could lead to the type of cherry-picking that the reviewer rightfully suggests avoiding. That said, it is well known that most lineages of marine prokaryoplankton, including *Pelagibacter*, do take up various organic compounds and incorporate them into their biomass. To address the reviewer's comment, we added this information and appropriate references to lines 216-219.

Referee #3 (Remarks to the Author):

I have reviewed the revised version of manuscript 2022-0202239A by Munson-McGee et al. I think the argument that *Pelagibacter*, SAR86, and perhaps other members of the prokaryoplankton use proteorhodopsin to produce ATP could be strengthened if the authors cited some papers that are currently missing from the reference list:

1. Béjà, O., L. Aravind, E. V. Koonin, M. T. Suzuki, A. Hadd, L. P. Nguyen, S. B. Jovanovich, C. Gates, R. A. Feldman, J. L. Spudich, E. N. Spudich, and E. F. DeLong. 2000. Bacterial rhodopsin: evidence for a new type of phototrophy in the sea. *Science* 289: 1902-1906.
2. Sabehi, G., Béjà, O., Suzuki, M.T., Preston, C.M. & E. F. DeLong. 2004. Different SAR86 subgroups harbour divergent proteorhodopsins, *Environ. Microbiol.* 6:903–910.
3. A. Martinez, A. Bradley, J. Waldbauer, R. Summons and E. F. DeLong. 2007.

Proteorhodopsin photosystem gene expression enables photophosphorylation in a heterologous host. Proc. Natl Acad Sci. 104: 5590-5595.

4. DeLong, E. F. and Béjà, O. (2010). The light-driven proton pump proteorhodopsin enhances bacterial survival during tough times. PLoS Biol 9: e1000359.

5. Dupont, C. L. et al. 2012. Genomic insights to SAR86, an abundant and uncultivated marine bacterial lineage. ISME Journal 6: 1186–1199.

I think these papers collectively make a very strong case that proteorhodopsin is involved in the production of ATP by some prokaryoplankton.

This was, to me, the only weak aspect of the original manuscript. The use of single-cell RSG fluorescence to estimate the respiration rates of single cells is certainly an exciting and potentially very important development. Scientific research is invariably methods limited to some extent, and this is a tool that will enable research that would have been impossible with any other methodology. I see no issues with the data & methodology or the use of statistics. I think the conclusions would be strengthened by citing the above references and information in those references.

We appreciate the reviewer's positive assessment of our work and their recognition of the applicability of our technique and the new research that it will enable. We have added the recommended references and spelled out some of their key findings in the manuscript. This provided valuable contextual information for the interpretation of our findings and helped addressing some of the comments made by other reviewers.

Referee #4 (Remarks to the Author):

The manuscript "Decoupling of respiration rates and abundance in marine prokaryoplankton" highlights results from an exciting new method designed to tackle a long standing problem, which is to quantify respiration rates in diverse lineages of uncultured marine bacteria. The justification for this work is well established and the authors do a nice job highlighting the significance of opening the black box of bulk respiration. The authors present a proteorhodopsin-based explanation to support their key observation, which is that the most abundant organisms in the ocean contribute very little to respiration and that rare organisms contribute to the bulk of respiration. Overall this is a very thorough and well written manuscript that introduces an exciting new method and that presents some new results. I have some concerns regarding the focus and some question regarding the method.

We appreciate the reviewers' overall positive assessment of the manuscript and the new method

Concern regarding the focus:

My primary concern is that the title and key observation highlighting the decoupling of key

parameters will be interpreted by some to mean that the most abundant organisms in the ocean are inactive (as noted by previous reviewers). An alternative explanation is that the most abundant lineages of bacteria in the ocean are also the most efficient and adjust carbon and energy sources to maintain high bacterial growth efficiency ($BGE = (BP)/(BP + BR)$), which is calculated with bacterial production not abundance. In their response to reviewers, the authors correctly note that the cells are not inactive, but that light-driven ATP production could provide a significant source of energy. There are other explanations that may contribute and I think that the authors should consider them as well. For example, the glyoxylate shunt, used by SAR11 and other aerobic heterotrophs, bypasses key CO₂ producing steps in the TCA cycle to metabolize specific compounds like fatty acids and two carbon compounds, and to regenerate key intermediates for biosynthesis. Also, in a relatively recent review of SAR11, Giovannoni notes that SAR11 do not use a wide range of energy rich carbohydrates. Having mechanisms that conserve carbon and energy, like the glyoxylate shunt and proteorhodopsin, makes sense in a carbon and energy limited systems, especially if the goal is to maintain high BGE. In short, suggesting that the dominant lineages don't contribute much to respiration will likely propagate the misleading notion that evolutionary success is achieved through dormancy. I encourage the authors to consider focusing on high and low BGE as an alternative.

This is an important comment, which relates to several comments by other reviewers. In response, we have added the following statement on lines 216-217: "While it is well-known that *Pelagibacter* and other predominant groups of surface ocean prokaryoplankton consume simple organic compounds, access to proteorhodopsin-derived ATP may increase the use of these molecules in biosynthesis as opposed to respiration, thus increasing growth efficiency". We also referred to growth efficiency in the Abstract. We believe that the title should be left as is, since we use the specific term "respiration rate" rather than "growth rate" or "metabolic rate".

Reservations regarding the focus:

Second but related, I don't think that decoupled abundance and respiration is surprising. The authors note that low respiration rates have been reported in cultured SAR11 (Steindler et al., 2011). Previous studies have also found that SAR11 don't use a wide range of energy rich carbohydrates but instead use compounds like VOCs, organic acids, phosphonates, polyamines, and amino acids (Giovannoni 2017). It has been known for decades that members of the Roseobacter are often the most abundant cells in phytoplankton blooms and members of this lineage are capable of using a broad range of energy rich carbohydrates. The authors point this out in the manuscript, noting that their data are in agreement with previous work. The ability to measure lineage-specific differences in respiration will fill a critical gap in knowledge and has great potential to transform our understanding of how different lineages have adapted to life in the oceans. But given what is already known about the metabolism of oligotrophs and copiotrophs, I think the novelty is in the development of the method and its application to field samples, not the decoupling of abundance and respiration.

We appreciate the reviewer's excitement for the development of the method as well as its application on field samples. We also agree that several prior studies have reported rates of growth and nutrient uptake by certain marine prokaryoplankton lineages. However, such prior studies are scarce, limited in phylogenetic scope, and often report contradictory findings, as we point out on lines 175-182. Most importantly, to the best of our knowledge, of the abundant prokaryoplankton lineages, respiration rates have been analyzed only in *Pelagibacter*, and only under laboratory conditions. This is a major gap in our understanding of the marine carbon cycle. Thus, we believe that the lineage-resolved, microbiome-wide, *in situ* respiration rates and their decoupling from lineage abundance, which we report in our manuscript, are highly novel and important to science.

Other significant findings:

Light driven ATP production in SAR11 and other oligotrophic marine bacteria is a potential source of energy, and to me a more interesting result. As the authors note, growth experiments have not shown that marine bacteria with proteorhodopsin have a significant advantage in the light versus the dark. As the authors have shown, respiration in light and dark samples might provide a better answer to how cells benefit, beyond what is known about the benefits associated with declining respiration when cells enter a non-growth state (Steindler et al., 2011). When more energy from the sun is available cells could respire less to maintain high BGE and to conserve carbon, but when less energy is available (no sun) cells could respire more, as the authors suggest. This section is a nice addition to the paper and represents a more novel discovery. Of course, doing this under controlled conditions with an isolate would be idea and although not feasible for this study, SAR11 isolates are available from several laboratories upon request.

We appreciate the reviewer's positive comment and agree that the implementation of our methods in experiments with SAR11 and SAR86 isolates is a great opportunity for future projects.

Questions about the method:

The organisms used to calibrate and validate the RSG-based prokaryoplankton respiration measurements all grow to very high cell densities and on nutrient rich media. They also consume large amounts of oxygen. I agree with other reviewers that this manuscript would be stronger if RSG-based respiration was measured in more representative bacteria and ideally under different growth conditions. In the primary RSG paper referenced by the authors (Kalyuznaya et al., 2008), they used starved cells to show a loss of RSG signal and then added substrate to show increases in RSG signal. In the present study, a loss of RSG signal is only shown for killed controls, which were autoclaved for 30 minutes. This seems a bit harsh. Experiments with representative cultures and with controls under different conditions or that have been starved and/or inactivated, rather than severely altered by autoclaving, would be more compelling. Ideally, it would be nice to have a range of relativized fluorescence intensities for representative organisms. As is, the authors are comparing respiration of low abundance slow growing cells in low nutrient seawater to high abundance fast growing cells in high nutrient media. More specific questions about this are below.

We agree with the reviewer that it is important to demonstrate the usability of the RSG approach with diverse microorganisms and under diverse conditions. To the extent that's practically possible, that is actually what we did, as indicated in Table S1 and Figure 1. One major consideration to keep in mind is that the traditional respiration rate assays rely on measurements of oxygen concentration changes over time. As we point out on lines 96-99 and 581-590, in order to detect such changes with either Winkler titration or optodes, it is necessary to run incubations for at least 24 h. Meanwhile, RSG incubations are only 30 minutes-long. This difference in incubation length makes it hard to compare the two techniques during the log phase, when both cell numbers and respiration rates keep changing. For this reason, we had to perform our respiration measurements during the stationary phase (lines 584-90), i.e., under nutrient-deplete conditions that may resemble the predominant conditions in the ocean to a larger extent than what the initial growth medium composition suggests. This is reflected in the successful replication in our cultures (Fig. 1A) of the range of cell-specific respiration rates that is found in the environment, including rates similar to those previously reported for *Pelagibacter* cultures (Figs. 2-4). Although we varied the initial growth medium composition for one of our cultures (Table S1), nutrient concentrations were likely similar by the time these two incubations entered the stationary phases and we started their respiration measurements. As a practical solution to the challenge of varying conditions at the stationary phase in a meaningful way, we varied incubation temperature (Fig. 1, Table S1).

In Figure 1 panel A, the authors plot normalized RSG fluorescence intensity against the rate of oxygen consumption, reported as respiration rate (fmol/cell/hour). In panel B, the authors plot RSG- and Winkler-based respiration measurements from natural samples, but with RSG-based respiration (umol oxygen/L/hour). I assume the difference in units is for direct comparison with the Winkler-based method but would like to know why. Why is panel A in fmol/cell/hour and panel B in umol oxygen/L/hour? Isn't relative fluorescence intensity used to calculate RSG-based respiration in panel B, ie. by using the correlation in panel A? If the points in pane B were added to panel A, using average relative fluorescence intensity values, where would they plot on the y axis (fmol/cell/hour)? If relative fluorescence intensity is low for the natural samples (0-20) then the per cell values are in the range of data that are only represented on part of the graph and that don't look correlated if the higher values obtained from non-marine organisms are removed. I looked for relative fluorescence intensity values in the tables but couldn't find them. Is oxygen consumption comparable between the cultures and the field samples?

In response to the reviewer's comment, we provided further clarifications in the legend of Figure 1B. Figures 1A and 1B serve different purposes, which is the reason for their different scales. On the one hand, the goal of Figure 1A is to show how we calibrated the RSG method, which performs respiration measurements on individual cells. The cell-specific scale used in 1A matches scales in Figures 2-4, all of which report cell-specific respiration rates in field samples. On the other hand, the goal of Figure 1B is to compare the RSG technique against the traditional, Winkler-based assays for the measurement of bulk respiration rates of marine prokaryoplankton. Here, we used a volume- rather than cell-specific scale, which is the most common way of reporting Winkler-based measurements. We believe that this makes it easier to compare our results to prior literature and offers an opportunity to evaluate

RSG method's suitability to perform bulk respiration measurements – a different application from what is reported in Figures 2-4, which is discussed on lines 83-102.

In Figure 2 panel D the y axis is “per cell respiration rate (fmol oxygen/hour)”. Is this the same as the y axis in Figure 1 panel A “respiration rate (fmol/cell/hour)”?

Yes, they are the same - we have changed the axis labels to be the same and therefore more clearly comparable.

And in Figure 4 panels C and D the y axis is “oxygen consumption amol/cell/hour.” Is this the same as respiration rate in Figure 1 panel A, per cell respiration rate in Figure 2 panel D, or are they all the same? Consistency would help with interpretation. Based on these other plots and discussions in the text, most natural samples have low relative fluorescence values.

These are also the same, and we have changed the labels accordingly. This also brought to our attention a typo that we had to fix (should be fmol instead of amol). Thank you for bringing this to our attention!

Reviewer Reports on the Second Revision:

Referees' comments:

Referee #1 (Remarks to the Author):

This version is substantially improved and the authors justify their responses to my review reasonably well. I am still a little disappointed with their depiction of prior studies. There are techniques such as FISH-CARD, Nano-SIMS that have been used to open the “black box” of microbial metabolism (Line 51). I'd like the authors to acknowledge this work that has already shown that many microbial cells are metabolically inactive or not very metabolically active. This is not a new finding. What is new is the level of detail and the measurement of respiration rates in particular, linked to taxonomy. The connection to expression of proteorhodopsin is also novel.

Line 208: In the discussion of mRNA and rRNA not being good proxies for cellular activity, I think a revision is necessary. Cell activity is many things, in this study only respiration was measured. So, mRNA and rRNA are not good proxies for respiration specifically. This is particularly important in light of the discussion starting on line 212 where the authors make the opposite argument – that the SAR11 could be very actively getting ATP from rhodopsin. In this case respiration is not needed for energy generation despite the possibility that the cells may still be growing or at least active.

In the discussion of diel cycles, the authors cite Ottesen which is great, but there are several others that point to differential metabolism over the diel, including diel cycles of transporters. One recent example is Muratore et al., 2022.

Once these issues are addressed I would support publication of this manuscript.

Referee #2 (Remarks to the Author):

The authors proposed an updated version of their manuscript. The points raised in the previous round of review have been satisfactorily addressed. I understood the metabolic modeling as being out of the scope and thank the authors for their clarification and data availability.

Referee #4 (Remarks to the Author):

The authors of "Decoupling of Respiration Rates and Abundance in Marine Prokaryoplankton" have done an excellent job addressing my comments and of improving the manuscript. I appreciate their efforts to provide clarity, specifically regarding abundance, respiration rates and growth efficiency. Based on their responses and changes, and my second review of the manuscript, I feel confident that this paper will make a significant contribution.

I still think that the emphasis on decoupling respiration rates from abundance (even if it is rates) will

be perceived by some as indicating that the most abundant organisms are not active. It seems that the primary conclusion (aside from the new method) is in Figure 2 C-E. Panel C is lineage abundance decreasing from left to right. Panel D is lineages respiration rate increasing from left to right. The opposite trends in C and D indicate decoupling. Panel E, however, is % lineage respiration and is more uniform, except for Roseobacteria and at one location in the ocean. For me, panel E has a more compelling conclusion because it gets at the importance of growth efficiency, which is particularly relevant considering the role of the marine carbon cycle in climate change. I am, however, satisfied with the improvements and additional efforts to clarify the main points.

Author Rebuttals to Second Revision:

Dear Dr. Caputa,

We are encouraged by the appreciation of the reviewers and their support for publication. In addition to addressing the reviewer comments, we have also reformatted the manuscript and supplemental materials to comply with the Nature formatting guide. This includes changing the order of sections within the manuscript file, removing the figures and extended data from the main text, and condensing the extended data down to 10 total figures. We also removed several sentences from the first paragraph of the main text, which were redundant to the Summary Paragraph.

Referee #1 (Remarks to the Author):

This version is substantially improved and the authors justify their responses to my review reasonably well. I am still a little disappointed with their depiction of prior studies. There are techniques such as FISH-CARD, Nano-SIMS that have been used to open the “black box” of microbial metabolism (Line 51). I’d like the authors to acknowledge this work that has already shown that many microbial cells are metabolically inactive or not very metabolically active. This is not a new finding. What is new is the level of detail and the measurement of respiration rates in particular, linked to taxonomy. The connection to expression of proteorhodopsin is also novel.

We thank the reviewer for their comment on the improvement of the manuscript. We agree that there are multiple techniques that have been used to investigate rates of prokaryoplankton growth and organic substrate uptake. However, our “black box” comment was specifically referencing microbial respiration, for which single cell techniques are much more limited. We have added content addressing several of the suggested techniques (lines 45-46).

Line 208: In the discussion of mRNA and rRNA not being good proxies for cellular activity, I think a revision is necessary. Cell activity is many things, in this study only respiration was measured. So, mRNA and rRNA are not good proxies for respiration specifically. This is particularly important in light of the discussion starting on line 212 where the authors make the opposite argument – that the SAR11 could be very actively getting ATP from rhodopsin. In this case respiration is not needed for energy generation despite the possibility that the cells may still be growing or at least active.

We appreciate the reviewers comment and suggestion. We have modified this section to clarify this point, lines 196-198: “This suggests that ribosomal and total mRNA molecule counts are poor indicators of cellular respiration and that energy producing processes other than respiration may be important to the metabolism of GoM prokaryoplankton.”

In the discussion of diel cycles, the authors cite Ottesen which is great, but there are several others that point to differential metabolism over the diel, including diel cycles of transporters. One recent example is Muratore et al., 2022.

Thank you for the suggestion. We have added a reference to Muratore et al. and additional discussion about diel cycles (lines 216-217): “Previous studies have demonstrated differential metabolism of prokaryoplankton between day and night, including a diurnal cycle of gene expression for transporters and proteorhodopsins”

Once these issues are addressed I would support publication of this manuscript.

Referee #2 (Remarks to the Author):

The authors proposed an updated version of their manuscript. The points raised in the previous round of review have been satisfactorily addressed. I understood the metabolic modeling as being out of the scope and thank the authors for their clarification and data availability.

We thank the reviewer for all of their time and comments on our manuscript. Their insight and suggestions have helped improve the manuscript.

Referee #4 (Remarks to the Author):

The authors of "Decoupling of Respiration Rates and Abundance in Marine Prokaryoplankton" have done an excellent job addressing my comments and of improving the manuscript. I appreciate their efforts to provide clarity, specifically regarding abundance, respiration rates and growth efficiency. Based on their responses and changes, and my second review of the manuscript, I feel confident that this paper will make a significant contribution.

I still think that the emphasis on decoupling respiration rates from abundance (even if it is rates) will be perceived by some as indicating that the most abundant organisms are not active. It seems that the primary conclusion (aside from the new method) is in Figure 2 C-E. Panel C is lineage abundance decreasing from left to right. Panel D is lineages respiration rate increasing from left to right. The opposite trends in C and D indicate decoupling. Panel E, however, is % lineage respiration and is more uniform, except for Roseobacteria and at one location in the ocean. For me, panel E has a more compelling conclusion because it gets at the importance of growth efficiency, which is particularly relevant considering the role of the marine carbon cycle in climate change. I am, however, satisfied with the improvements and additional efforts to clarify the main points.

We appreciate the reviewer's positive assessment of our edits and their response to the revised manuscript. To further emphasize the distinction between respiration and growth, we have slightly modified sentences on lines 37-39 and 196-198: "These findings suggest that prokaryoplankton's dependence on respiration and remineralization of phytoplankton-derived organic carbon into CO₂ for its energy demands and growth may be lower than commonly assumed and variable among lineages."; "This suggests that ribosomal and total mRNA molecule counts are poor indicators of cellular respiration and that energy producing processes other than respiration may be important to the metabolism of GoM prokaryoplankton."

Reviewer Reports on the Third Revision:

Referees' comments:

Referee #1 (Remarks to the Author):

Line 224 prokaryotic phototrophs include cyanobacteria but I think the authors here mean classic heterotrophic bacteria that use proteorhodopsin. Please clarify. Maybe they could be called non-cyanobacterial prokaryotic phototrophs? Similar to the language for non-cyanobacterial diazotrophs.

Line 226 Does UBA10364 (Bacteroidota, Flavobacteriales) contain proteorhodopsin? I assume it does not but it might be good to state this to emphasize it.

Line 240 Add "a" to: Impact of "a" proteorhodopsin, or make pump plural.

Line 257 the last line of the conclusions is both indirect/vague and also a bit grandiose claiming widespread implications for C and energy cycling. I think that this paper is about the idea that some prokaryotes that we classically think of as heterotrophs may actually be using pathways other than the TCA cycle to gain energy, even when the TCA cycle appears to be their central metabolism. This has implications for how the cell functions and grows. So my original question remains unanswered – where is the carbon and nitrogen coming from for growth and maintenance if these bacteria are fueled by ATP production through sunlight. In a chlorophyll based organism they would be capturing energy from sunlight and using it to fix carbon using rubisco. Then they respire that sugar in the TCA cycle to provide substrates for growth and ATP. So what is the proteorhodopsin based ATP production coupled to that could allow for growth or any impact on the carbon or nitrogen cycle? Transport of organic matter into the cell? Carbon fixation? Ammonia/nitrate uptake? If the authors don't know they should say they don't know. If they have hypotheses they should state those. Many exist in the literature. As it stands I learn from this manuscript that some key bacteria get energy from rhodopsin (which I knew) but that they do not respire (which seems paradoxical if you are claiming that the rhodopsin is somehow allowing them to grow as opposed to just allowing them to remain alive with some basal metabolism but not grow). Alone, the proton pump provided by rhodopsin does not allow for growth. In fact you still need the building blocks for the ATP to be produced in some way. Where does this come from, or is it just phosphorylated and dephosphorylating (ADP-ATP cycling)? I think it is possible that the ATP can be driving many biosynthetic pathways directly while bypassing the TCA cycle, but this implies the availability of some suite of substrates from the environment. These substrates are likely more available during the day when phytoplankton are active but could also come during viral lysis or grazing, etc. Citation #35 (Martinez) for example discusses these ideas as do many other citations. What does light generated ATP production do for the cells that have rhodopsin? I don't expect the authors to answer this but only to discuss what is known in the literature about the function of energy produced through rhodopsins and how a cell could grow better with this energy than without it. The authors may think this is so obvious it is not worth mentioning or outside the scope of the work here, but I don't think it is.

Referee #4 (Remarks to the Author):

The author's have addressed all of my prior comments and the slight changes to sentences regarding respiration and growth are appropriate.

Author Rebuttals to Third Revision:

Response to Reviewers

Referee #1 (Remarks to the Author):

Line 224 prokaryotic phototrophs include cyanobacteria but I think the authors here mean classic heterotrophic bacteria that use proteorhodopsin. Please clarify. Maybe they could be called non-cyanobacterial prokaryotic phototrophs? Similar to the language for non-cyanobacterial diazotrophs.

We agree and have clarified the language to specify non-cyanobacterial prokaryotic phototrophs, lines 227-229.

Line 226 Does UBA10364 (Bacteroidota, Flavobacterales) contain proteorhodopsin? I assume it does not but it might be good to state this to emphasize it.

We did detect proteorhodopsin genes in UBA10364 (Supplemental Table 4). We found proteorhodopsin genes in most genera with >5 cells (91/111). However, our results indicate that these genes may play somewhat different roles among lineages. For example, many of the Rhodobacteraceae contain proteorhodopsin genes, but the expression level of these genes is low, while cell-specific respiration rates are high.

Line 240 Add “a” to: Impact of “a” proteorhodopsin, or make pump plural.

Pump is now plural

Line 257 the last line of the conclusions is both indirect/vague and also a bit grandiose claiming widespread implications for C and energy cycling. I think that this paper is about the idea that some prokaryotes that we classically think of as heterotrophs may actually be using pathways other than the TCA cycle to gain energy, even when the TCA cycle appears to be their central metabolism. This has implications for how the cell functions and grows. So my original question remains unanswered – where is the carbon and nitrogen coming from for growth and maintenance if these bacteria are fueled by ATP production through sunlight. In a chlorophyll based organism they would be capturing energy from sunlight and using it to fix carbon using rubisco. Then they respire that sugar in the TCA cycle to provide substrates for growth and ATP. So what is the proteorhodopsin based ATP production coupled to that could allow for growth or any impact on the carbon or nitrogen cycle? Transport of organic matter into the cell? Carbon fixation? Ammonia/nitrate uptake? If the authors don't know they should say they don't know. If they have hypotheses they should state those. Many exist in the literature. As it stands I learn from this manuscript that some key bacteria get energy from rhodopsin (which I knew) but that they do not respire (which seems paradoxical if you are claiming that the rhodopsin is somehow allowing them to grow as opposed to just allowing them to remain alive with some basal metabolism but not grow). Alone, the proton pump provided by rhodopsin does not allow for growth. In fact you still need the building blocks for the ATP to be produced in some way. Where does this come from, or is it just phosphorylated and dephosphorylating (ADP-ATP cycling)? I think it is

possible that the ATP can be driving many biosynthetic pathways directly while bypassing the TCA cycle, but this implies the availability of some suite of substrates from the environment.

These substrates are likely more available during the day when phytoplankton are active but could also come during viral lysis or grazing, etc. Citation #35 (Martinez) for example discusses these ideas as do many other citations. What does light generated ATP production do for the cells that have rhodopsin? I don't expect the authors to answer this but only to discuss what is known in the literature about the function of energy produced through rhodopsins and how a cell could grow better with this energy than without it. The authors may think this is so obvious it is not worth mentioning or outside the scope of the work here, but I don't think it is.

We thank the reviewer for their comments and have revised the last lines to address these points.

Referee #4 (Remarks to the Author):

The author's have addressed all of my prior comments and the slight changes to sentences regarding respiration and growth are appropriate.

We thank the reviewer for their insightful comments and support for publication.